# FLAG: Foundation model representation with Latent diffusion Alignment via Graph for spatial gene expression prediction

Qi Si [* 1]   Penglei Wang [* 2]   Yushuai Wu [1]   Yifeng Jiao [1]   Xuyang Liu [1]   Xin Guo [3 1]   Yuan Qi [1 3 4]   Yuan Cheng [3 1]

## Abstract

Predicting spatial gene expression from routine H&E enables large-scale molecular profiling, yet current models treat this as isolated pointwise tasks, thereby overlooking essential biological structures like gene coordination and spatial distribution. To preserve these relationships, we introduce **FLAG**, a diffusion-based framework that redefines this task as structured distribution modeling. At the same time, we identify the critical **Gene Dimension Curse**, where joint modeling gene expression and their spatial interactions fail in high-dimensional spaces, and FLAG solves this challenge by integrating a spatial graph encoder for topological consistency and utilizing Gene Foundation Model (GFM) alignment for gene-gene fidelity in the generation process. To rigorously assess model performance, we propose a set of novel structural evaluation metrics, including Gene Structural Correlation (**GSC**) and Spatial Structural Correlation (**SSC**). Our experiments demonstrate that FLAG is highly competitive in traditional accuracy (PCC/MSE) while achieving significantly enhanced structural fidelity in capturing both gene-gene and gene-spatial relationships. The code is available at https://github.com/darkflash03/FLAG.

## 1. Introduction

Spatial transcriptomics (ST) measures gene expression while preserving spatial organization, enabling analyses of tissue microenvironments, multicellular programs, and disease niches (Ståhl et al., 2016; Rodriques et al., 2019). Although ST sequencing is expensive and low-throughput, H&E whole-slide images (WSIs) are readily available in all clinical workflows. This has motivated increasing interest in predicting spatial gene expression directly from WSIs.

Despite progress in discriminative methods, existing histology-to-spatial gene expression models overwhelmingly treat gene inference as independent scalar regressions, evaluated by pointwise metrics such as MSE or PCC. However, these metrics ignore gene–gene relationships underlying regulatory programs (Subramanian et al., 2005), as well as gene-spatial correlation that reflects tissue architecture (Dries et al., 2021). These structural properties are often more biologically informative than absolute expression values, which are essential for pathway analysis, spatial domain discovery, and microenvironment characterization. As a result, current models may achieve high per-gene accuracy yet produce gene expression maps that lack coherent internal structure.

To explicitly evaluate these structural aspects, we introduce two metrics: Gene Structural Correlation (GSC) and Spatial Structural Correlation (SSC). GSC evaluates the preservation ability of regulatory integrity by assessing whether the model maintains the coordinated interaction patterns inherent in gene-gene networks, which is essential for downstream pathway analysis. SSC assesses spatial distribution of genes by comparing predicted and true Moran's I (Moran, 1950) for each gene, capturing whether spatial organization is maintained to support downstream spatial domain identification.

Once the goal is reframed from predicting isolated values to preserving biological validity, the modeling objective naturally shifts from deterministic regression to structured distribution modeling. Crucially, the mapping from histology to gene expression is often stochastic and one-to-many, but regression methods tend to average out these variations, resulting in over-smoothed predictions. Diffusion models (Ho et al., 2020) provide a powerful solution for this task. By explicitly learning to approximate the high-dimensional probability manifold rather than just the conditional expectation, they preserve the intrinsic variance and joint correlations that pointwise objectives inherently ignore.

---

[1]Shanghai Academy of Artificial Intelligence for Science, Shanghai, China. [2]School of Biomedical Engineering, Shanghai Jiao Tong University, Shanghai, China. [3]Artificial Intelligence Innovation and Incubation Institute, Fudan University, Shanghai, China. [4]Zhongshan Hospital, Fudan University, Shanghai, China. Correspondence to: Xin Guo <guoxin@sais.org.cn>, Yuan Cheng <cheng_yuan@fudan.edu.cn>.

*Proceedings of the 43rd International Conference on Machine Learning*, Seoul, South Korea. PMLR 306, 2026. Copyright 2026 by the author(s).

To capture such structural dependencies, a reasonable strategy is the graph-diffusion approach (Jo et al., 2022), treating spots as nodes and relationships between nodes as edges to simultaneously diffuse expression and topological patterns. However, we identify a critical gene dimension curse through systematic exploration: while joint node–edge diffusion is effective for small gene sets, its optimization stability deteriorates rapidly as gene dimensionality increases. This finding directly motivates the architecture of FLAG.

Specifically, FLAG introduces a novel Spatial Graph Encoder to capture relationships between spots (Dries et al., 2021), providing spatial embeddings as conditioning signals. These signals then guide a gene-level diffusion process, effectively transforming the joint node-edge diffusion challenge into a conditional generation task. To further ensure biological fidelity, where the limited spots in ST slides constrain the statistical power to infer reliable gene regulatory relationships, we align our model with pretrained embeddings from GFM (Theodoris et al., 2023; Cui et al., 2024). By unifying these components, FLAG offers a principled framework for predicting high-dimensional spatial transcriptomics with superior structural fidelity. Our main contributions are:

- **Gene Dimension Curse analysis**: We systematically characterize how graph-diffusion behavior varies across gene dimensionalities and provide theoretical insight into why joint node–edge diffusion becomes unstable in high-dimensional settings.

- **FLAG framework**: We propose a structure-aware diffusion framework that leverages a spatial graph encoder and GFM-aligned gene representations to better match the spatial distribution and gene-level dependency structure.

- **Comprehensive evaluation**: We introduce additional metrics that explicitly assess biological structure, including GSC and SSC, providing a more meaningful assessment of generative fidelity than pointwise accuracy alone. Across multiple datasets, our method achieved significant improvements in these structural metrics while maintaining strong competitiveness in traditional pointwise metrics (PCC/MSE).

## 2. Related Work

**Diffusion Models for ST Predictions**  Diffusion models have emerged as powerful generative frameworks for learning complex high-dimensional data distributions. Foundational work such as DDPM (Ho et al., 2020), score-based generative modeling via stochastic differential equations (Song et al., 2021), and latent diffusion (Rombach et al., 2022) established that diffusion processes can capture rich cross-dimensional correlations and scale effectively to structured outputs. Motivated by these advances, SpaDiT (Li et al., 2024b) uses a diffusion transformer to predict ST expression from scRNA-seq, stDiff (Li et al., 2024a) formulates ST imputation as a diffusion-guided reconstruction problem, and SpotDiffusion (Chen et al., 2025) integrates scRNA-seq guidance to improve imputation of spot-level gene expression, while DiffusionST (Cui et al., 2025) uses diffusion-based denoising to enhance spatial expression profiles and sharpen domain boundaries. Only recently have generative diffusion frameworks been introduced for WSI-to-ST prediction, Stem (Zhu et al., 2025b) models spot-wise gene expression via conditional diffusion on histology-derived features, and STFlow (Huang et al., 2025) employs flow matching to synthesize expression fields. However, these methods do not fully exploit the spatial neighborhood context or leverage GFM.

**Gene Foundation Models and Representation Alignment**
Geneformer (Theodoris et al., 2023) introduced one of the first large-scale transformers trained on tens of millions of single cells to learn gene-level embeddings reflecting regulatory and pathway structure. Building on larger datasets, scGPT (Cui et al., 2024) learned joint gene–cell embeddings via generative pretraining on over 33 million profiles, while CellPLM (Wen et al., 2024) further extended cell-level embedding pretraining with masked expression modeling. More recent models scale to even larger corpora: scFoundation (Hao et al., 2024) trained a 100M-parameter transformer to obtain transferable gene and cell representations, and CellFM (Zeng et al., 2025) expanded to 100 million cells with improved modeling of gene–gene and cell–cell relationships. These models collectively indicate that pretrained gene/cell embeddings encode rich functional structure that cannot be recovered from limited ST datasets alone. Parallel to these biological models, the vision and generative modeling communities have explored aligning diffusion models with pretrained representation spaces. REPA (Yu et al., 2025) showed that aligning diffusion states with frozen image encoders (e.g., DINO (Oquab et al., 2024; Siméoni et al., 2025)) improves generative stability and semantic coherence, while SVG (Shi et al., 2026) demonstrated that diffusion trained directly in semantically structured frozen latents yields faster convergence and higher fidelity. More related works are detailed in Appendix A.

## 3. Method

### 3.1. Problem Formulation

Given a histology whole-slide image (WSI), we consider $S$ spatial measurement spots. Each spot $s \in \{1, \ldots, S\}$ is associated with: a 2D tissue coordinate $u_s \in \mathbb{R}^2$, a histology patch encoded into a visual feature $v_s \in \mathbb{R}^{d_v}$

using pretrained WSI encoders (Lu et al., 2023), and a high-dimensional gene expression vector $x_s \in \mathbb{R}^G$. We define a graph $\mathcal{G} = (\mathcal{V}, \mathcal{E})$ over spots, where nodes correspond to spots information and edges reflect local microenvironmental relations. Prior studies show that spot relationship arises from three primary factors: (1) physical proximity in the tissue coordinate space; (2) histology similarity derived from WSI features; (3) molecular similarity in expression space (Zhang et al., 2025). Given the WSI-derived visual features $\{v_s\}$, coordinates $\{u_s\}$, and graph $\mathcal{G}$, our goal is to predict the full set of spot-level gene expression vectors $\{x_s\}_{s=1}^S$. More method preliminaries can be seen in Appendix B.

### 3.2. Motivating Attempt: Joint Node–Edge Diffusion

As outlined in Preliminaries, the identity of a spot is defined not just by its own histology, but also by its relationships with neighbors. Although physical distance and histological similarity are directly observable, the most informative transcription-related signals remain latent during the inference process. To assess the potential value of recovering latent functional edges, we conduct a pilot study on a subset of High-Mean and High-Variance Genes (HMHVG) ($G = 50$). We compare graph-based predictors using only observable edges (physical distance and image similarity) against an Oracle setting in which ground-truth transcriptional correlations are provided. As shown in Fig. 1, access to Oracle correlation edges yields a substantial performance improvement. This result suggests that latent functional topology provides a strong constraint for spatial gene prediction, motivating explicit modeling of spot-spot correlations.

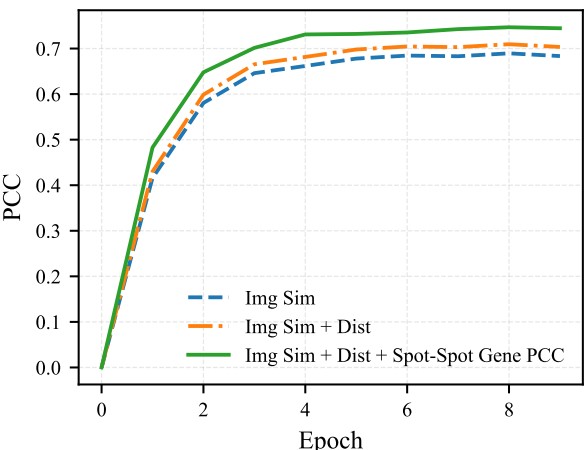

*Figure 1.* **Ablation study on edge attributes (HEST-1K (Jaume et al., 2024) HER2ST Dataset).** Comparison of PCC performance using different edge construction strategies: Image Similarity only, Image Similarity + Distance, and with additional Spot-Spot Gene PCC.

**Joint Node-Edge Diffusion** To break the circular dependency between expression and correlations, we treat both node states and functional edges as generative targets and model the joint distribution $p(\mathbf{X}, \mathbf{A} \mid \mathcal{C})$ rather than only $p(\mathbf{X} \mid \mathcal{C})$, where $\mathbf{A}$ denotes latent correlation edges generated from gene expression. We instantiate this via Graph Diffusion to learn $p(\mathbf{X}, \mathbf{A} \mid \mathcal{C})$ on a fully connected tissue graph $\mathcal{G} = (\mathcal{V}, \mathcal{E})$: nodes are $\mathbf{X} \in \mathbb{R}^{N \times G}$, the spot-wise gene expression; edges are $\mathbf{A} \in \mathbb{R}^{N \times N}$, the functional connectivity with $a_{ij} = \mathrm{corr}(\mathbf{x}_i, \mathbf{x}_j)$; conditioning is $\mathcal{C} = (\mathbf{C}_v, \mathbf{C}_e)$, where $\mathbf{C}_v \in \mathbb{R}^{N \times d_v}$ are H&E spot image features from a pretrained encoder and $\mathbf{C}_e \in \mathbb{R}^{N \times N \times 2}$ with $\mathbf{C}_{e,ij} = [d_{ij}, s_{ij}]$ concatenating physical distance and visual similarity.

**Backbone** To parameterize the joint score function $s_\theta(\mathbf{X}_t, \mathbf{A}_t, t, \mathcal{C})$, we develop a Graph Transformer that actively fuses the evolving graph structure with static conditions. Specifically, we modified the standard self-attention mechanism and proposed a novel **Edge-Modulated Attention** that enforces topological structures. Let $\mathbf{q}_i$ and $\mathbf{k}_j$ be the standard linear query and key projections of the node features $\mathbf{X}_t$. The attention matrix is explicitly gated by the noisy edge state $\mathbf{A}_t$ and the static edge condition $\mathbf{C}_e$, allowing the generated functional connections to dynamically strengthen message passing during the denoising process. The attention score $\mathbf{S}_{ij}$ is computed as:

$$\mathbf{S}_{ij} = \left( \frac{\mathbf{q}_i \mathbf{k}_j^\top}{\sqrt{d}} \right) \odot \underbrace{\left( 1 + \mathrm{Linear}(\mathbf{A}_{t,ij}) + \alpha \, \mathrm{Linear}(\mathbf{C}_{e,ij}) \right)}_{\text{Structural Gating}}$$
$$+ \underbrace{\left( \mathrm{Linear}(\mathbf{A}_{t,ij}) + \gamma \, \mathrm{Linear}(\mathbf{C}_{e,ij}) \right)}_{\text{Structural Bias}}.$$

$$(1)$$

where $\odot$ denotes element-wise multiplication, the Linear function denotes a scalar affine projection, and $\alpha, \gamma$ are learnable scalars. Conceptually, Structural Gating acts as a multiplicative modulator to scale the attention magnitude, whereas Structural Bias acts as an additive shift applied to the attention logits.

Crucially, to bridge this mechanism with the generative objective, the attention matrix $\mathbf{S}$ serves as the core topology to jointly update both the node and edge hidden states (detailed in Appendix D.2). Subsequently, these updated states are projected to the joint score components $s_\theta^{(X)}$ and $s_\theta^{(A)}$ required for the denoising loss.

**Training Objective** We train the joint node-edge diffusion model using a composite objective. The basic loss is the joint graph score-matching objective $\mathcal{L}_{\mathrm{graph}}$ (Eq. (15)), which supervises the denoising of both node states and edge states. While edges $\mathbf{A}$ are explicitly generated during diffusion, they are not semantically independent variables: by definition, functional edges are induced by the underlying gene expression. To enforce this dependency, we introduce

a **consistency loss** that aligns the generated edge state with the correlations implied by the generated node state at each denoising step:

$$\mathcal{L}_{\text{cons}} = \mathbb{E}_t \left\| \hat{\mathbf{A}}_0(\mathbf{A}_t) - \text{Corr}\left(\hat{\mathbf{X}}_0(\mathbf{X}_t)\right) \right\|_1. \quad (2)$$

The final training objective is

$$\mathcal{L} = \mathcal{L}_{\text{graph}} + \lambda_c \, \mathcal{L}_{\text{cons}}. \quad (3)$$

This constraint ensures that despite edge correlations being generated as latent variables, they remain consistent with the functional relationships implicitly defined by denoised gene expression, thereby preventing degenerate or inconsistent graph structures. Implementation details of graph diffusion are provided in Appendix D. This dynamic joint formulation serves as our exploratory benchmark to probe the ability of explicitly modeling transcriptional co-expression relationships.

### 3.3. Gene Dimension Curse

As the gene panel size $G$ increases, performance degrades for all methods due to the higher dimensional prediction difficulty. However, we observe a markedly steeper scaling behavior for joint node-edge diffusion than node-only diffusion. We systematically evaluated the model by varying the gene panel size $G$. As illustrated in Figure 2(a), we observe a distinct phase transition governed by dimensionality: as gene size grows, the performance of joint diffusion deteriorates rapidly and eventually collapses.

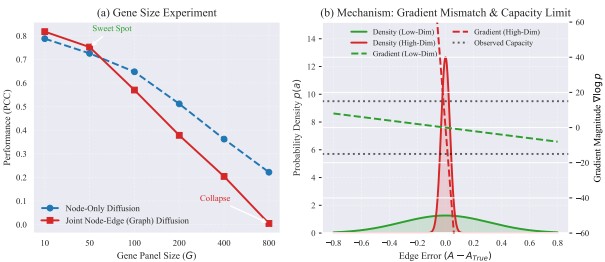

*Figure 2.* **Failure of joint node-edge diffusion. (a) Empirical performance on HER2ST as gene dimensionality $G$ increases.** Joint node-edge diffusion performs well at small $G$ but collapses beyond a critical dimensionality, while node-only diffusion degrades more smoothly. **(b) Mechanism analysis.** As $G$ increases, simulated correlation distribution concentrate sharply, causing the consistency manifold to become increasingly narrow. This induces high-curvature score functions with gradient magnitudes that exceed the limited effective capacity of the neural network, leading to unstable optimization and training collapse.

**Why Collapse?** The failure arises from how correlation estimation behaves in high dimensions. Figure. 2(b) provides an intuitive geometric explanation. When the gene dimension $G$ is small, empirical correlation estimates between

spots exhibit high variance. As a result, the set of consistent node-edge configurations $\{(\mathbf{X}, \mathbf{A}) : \mathbf{A} = \text{corr}(\mathbf{X})\}$ forms a thick manifold. In this regime, the conditional score $\nabla_{\mathbf{A}_t} \log p(\mathbf{A}_t \mid \mathbf{X}_t)$ varies smoothly and can be reliably approximated by a neural network, allowing joint diffusion to act as an effective spatial regularizer. In contrast, as the gene dimension increases, correlation estimates concentrate sharply around their population values. This concentration causes the consistency manifold to become increasingly thin, meaning that only a very narrow range of edge configurations is compatible with a given node state. To enforce consistency, the diffusion model must approximate a highly curved score field with large gradient magnitudes. These gradients grow with gene dimension and rapidly exceed the effective approximation capacity of finite-width networks, leading to unstable optimization and training collapse.

**Formulation** We formalize this limitation as an optimization lower bound. As $G$ increases, empirical correlations concentrate sharply, forcing the network to approximate a highly curved score field. Let $\mathcal{L}^*_{\text{joint}}$ and $\mathcal{L}^*_{\text{node}}$ denote the minimum achievable training losses of joint node-edge diffusion and node-only diffusion, respectively. The optimization gap satisfies:

$$\mathcal{L}^*_{\text{joint}}(G) - \mathcal{L}^*_{\text{node}} \geq \Omega(G). \quad (4)$$

This bound mathematically explains our empirical observations: When gene sizes are small, the joint diffusion converges effectively and successfully captures co-expression structures. However, the $\Omega(G)$ optimization penalty triggers a fundamental collapse in high-dimensional patterns. The full derivation is provided in Appendix E. To achieve a generalized solution that scales robustly to comprehensive gene panels, we must overcome this theoretical bottleneck.

### 3.4. From Joint Node-Edge Diffusion to Spatial Conditioning

Our Gene Dimension Curse analysis identifies a fundamental barrier in high-dimensional ST generation. While Joint Node-Edge diffusion suffers a catastrophic collapse due to variance explosion (Fig. 2(a), Red Line), we observe that Node-Only Graph Diffusion is also insufficient. As shown in Fig. 2(a) (Blue Line), the performance of node-only methods drops significantly as the gene dimension $G$ increases (from PCC > 0.8 to ~ 0.2). In contrast, the diffusion models that condition only on image features (Stem (Zhu et al., 2025b)) achieve better numerically stable as $G$ increases, but they discard explicit spot-spot structure and without using spatial context.

These observations motivate a different use of the graph: rather than diffusing on the graph, we use it as a spatial encoder. We fix the topology using reliable priors and apply

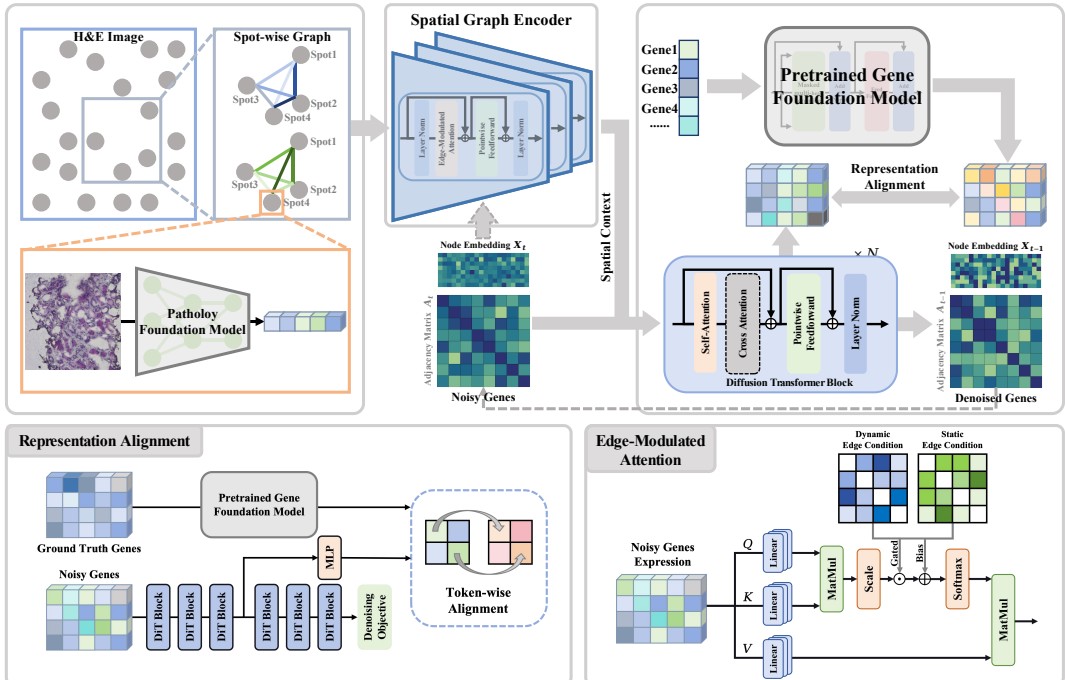

*Figure 3.* **The FLAG Framework Architecture.** Left: H&E tiles are encoded by a pathology foundation model and assembled into a spot-wise graph, which a graph encoder aggregates into spatial context embeddings $\mathbf{H}_{\text{spatial}}$. Right: a conditional diffusion transformer denoises noisy gene expression $X_t$ under this spatial context, while an intermediate-layer alignment constrains hidden states to match embeddings from a pretrained Gene Foundation Model, jointly enforcing spatial and gene-structural coherence.

a graph encoder to aggregate neighborhood information,

$$\mathbf{H}_{\text{spatial}} = \text{GraphEncoder}\big(\mathbf{C}_v, \mathbf{C}_e\big), \qquad (5)$$

where $\mathbf{C}_v$ and $\mathbf{C}_e$ are the node and edge conditions. The graph no longer carries a high-dimensional generative target, but produces a compact per-spot spatial context $\mathbf{H}_{\text{spatial}}$ that is stable with respect to $G$. The actual generation is then performed in gene space: a gene-wise diffusion backbone denoises $\mathbf{X}_t$ conditioned on $\mathbf{H}_{\text{spatial}}$,

$$\hat{\epsilon} = \epsilon_\theta\big(\mathbf{X}_t \mid \mathbf{H}_{\text{spatial}}, t\big), \qquad (6)$$

so that spatial structure as a conditioning signal rather than an additional high-dimensional output. From the perspective of attention, a graph diffusion model mainly performs spot-to-spot message passing: each $G$-dimensional expression vector is treated as a single feature and gene-gene relations are only implicitly modeled. The image-conditioned diffusion model such as Stem performs gene-to-gene attention within each spot but ignores interactions between neighboring spots. Our factorization combines the strengths of both: the graph encoder captures spot-spot interactions once in $\mathbf{H}_{\text{spatial}}$, while the gene diffusion backbone focuses on gene-gene dependencies conditioned on the spatial context.

## 3.5. Gene Foundation Model (GFM) Alignment

The above analysis focuses on spatial structure: we use a graph backbone to encode spot-spot relationships and condition gene diffusion on rich spatial context. However, high-quality ST generation also requires preserving gene-gene relationships. Previous diffusion (e.g., Stem) infers these relationships from a few thousand ST slides with limited gene coverage, which hinders the effective estimation of the learned covariance. To inject robust gene semantics, we align our gene diffusion backbone with large-scale GFM.

**Graph-Conditioned Gene Diffusion Backbone.** At each diffusion step $t$, the model denoises a noisy gene expression matrix $\mathbf{X}_t \in \mathbb{R}^{B \times G}$ for $B$ spots and $G$ genes, conditioned on the spatial context vectors $\mathbf{C}_{\text{graph}}$ produced by the graph encoder:

$$\epsilon_\theta(\mathbf{X}_t) = \text{DiT}(\mathbf{X}_t \mid \mathbf{C}_{\text{graph}}, t). \qquad (7)$$

Here DiT is a latent diffusion transformer operating along the gene dimension.

**GFM Alignment Loss.** Let $\mathbf{F} \in \mathbb{R}^{G \times d_e}$ denote fixed per-gene embeddings extracted offline from a pretrained GFM. Details of the preprocessing pipeline for obtaining $F$ are

*Table 1.* **Quantitative results on the Top-200 HMHVG panel across HEST-1k datasets.** (D) and (G) denote discriminative and generative methods, respectively. Performance is reported as Mean $\pm$ Std across test slides.

| Method | PCC $\uparrow$ | MSE $\downarrow$ | GSC $\uparrow$ | SSC $\uparrow$ |
|---|---|---|---|---|
| **HER2ST Dataset** | | | | |
| HisToGene (D) | 0.4940$\pm$0.0032 | 1.8459$\pm$0.0286 | 0.2065$\pm$0.0131 | -0.1549$\pm$0.0436 |
| BLEEP (D) | 0.4852$\pm$0.0012 | 1.1516$\pm$0.0158 | 0.6988$\pm$0.0246 | 0.1886$\pm$0.0384 |
| TRIPLEX (D) | 0.6913$\pm$0.0024 | **0.6559$\pm$0.0214** | 0.5593$\pm$0.0154 | 0.0708$\pm$0.0264 |
| Stem (G) | 0.5772$\pm$0.0084 | 0.9535$\pm$0.0142 | 0.8322$\pm$0.0084 | 0.3810$\pm$0.0195 |
| STFlow (G) | **0.7058$\pm$0.0052** | 0.6769$\pm$0.0138 | 0.7890$\pm$0.0174 | 0.2890$\pm$0.0263 |
| **FLAG (Ours)** | 0.6835$\pm$0.0084 | 0.7342$\pm$0.0156 | **0.8926$\pm$0.0106** | **0.6386$\pm$0.0184** |
| **KIDNEY Dataset** | | | | |
| HisToGene (D) | 0.2318$\pm$0.0023 | 1.3935$\pm$0.0452 | -0.0264$\pm$0.0284 | 0.1696$\pm$0.0349 |
| BLEEP (D) | 0.1471$\pm$0.0047 | 2.7602$\pm$0.0819 | 0.5331$\pm$0.0731 | 0.0889$\pm$0.0876 |
| TRIPLEX (D) | 0.3739$\pm$0.0089 | **1.1454$\pm$0.0137** | 0.4469$\pm$0.0592 | 0.2347$\pm$0.0512 |
| Stem (G) | 0.3443$\pm$0.0012 | 1.3828$\pm$0.0904 | 0.8451$\pm$0.0108 | 0.1257$\pm$0.0198 |
| STFlow (G) | 0.3145$\pm$0.0065 | 1.2790$\pm$0.0628 | 0.6857$\pm$0.0846 | -0.1007$\pm$0.0765 |
| **FLAG (Ours)** | **0.3917$\pm$0.0074** | 1.2112$\pm$0.0375 | **0.8713$\pm$0.0415** | **0.3409$\pm$0.0423** |
| **PRAD Dataset** | | | | |
| HisToGene (D) | 0.2553$\pm$0.0054 | 1.9681$\pm$0.0842 | 0.3810$\pm$0.0521 | -0.1277$\pm$0.0275 |
| BLEEP (D) | 0.0417$\pm$0.0012 | 5.6868$\pm$0.0153 | 0.4338$\pm$0.0984 | 0.2897$\pm$0.0891 |
| TRIPLEX (D) | 0.5267$\pm$0.0087 | 1.5824$\pm$0.0981 | 0.5533$\pm$0.0315 | 0.6343$\pm$0.0442 |
| Stem (G) | 0.4025$\pm$0.0045 | 2.0938$\pm$0.0420 | 0.8216$\pm$0.0762 | 0.2768$\pm$0.0619 |
| STFlow (G) | 0.5337$\pm$0.0093 | 1.8776$\pm$0.0675 | 0.7228$\pm$0.0198 | 0.5638$\pm$0.0134 |
| **FLAG (Ours)** | **0.5853$\pm$0.0029** | **1.3771$\pm$0.0239** | **0.8775$\pm$0.0476** | **0.7510$\pm$0.0988** |

provided in Appendix F. These embeddings are not used as inputs at inference time; instead, they act purely as a gene structural prior during training.

Given the hidden representation $\mathbf{H}^{(k)} \in \mathbb{R}^{B \times G \times d_h}$ at an intermediate DiT block $k$, we align it to the GFM manifold via

$$\mathcal{L}_{\text{align}} = -\frac{\langle \text{MLP}(\mathbf{H}^{(k)}), \mathbf{F} \rangle}{\|\text{MLP}(\mathbf{H}^{(k)})\|_2 \|\mathbf{F}\|_2 + \epsilon}. \quad (8)$$

### 3.6. Overall FLAG Framework

By integrating functional components, FLAG couples a spatial graph encoder with a gene-level diffusion transformer, and regularizes the latter with a GFM. The full architecture is illustrated in Figure. 3.

To resolve the scalability bottleneck identified in Sec.3.3, FLAG transitions from the joint node-edge generation explored to a spatial-conditioning paradigm. Unlike the motivating attempt where the graph $\mathbf{A}$ was a sampled latent variable, the final FLAG treats the tissue topology as a fixed deterministic prior $\mathbf{C}_e$, which is derived from spatial and histological proximity.

This design ensures that the diffusion backbone focuses on capturing the joint data distribution of expression levels, guided by $\mathbf{H}_{\text{spatial}}$ as a conditioning signal. To further align generations with global biology, we impose a GFM

alignment loss on intermediate DiT representations. The overall training objective adds this gene-level alignment to the standard score-matching loss (Eq. 12):

$$\mathcal{L}_{\text{total}} = \mathcal{L}_{\text{score}} + \lambda_{\text{align}} \mathcal{L}_{\text{align}}. \quad (9)$$

In summary, the final FLAG fuses spot-spot structure (via the static graph encoder) with gene-gene structure (via GFM alignment). This factorization allows the model to leverage rich spatial and biological priors without suffering from the high-dimensional optimization penalty, yielding spatially and biologically coherent expression fields. Detailed algorithms are provided in Appendix G.

## 4. Experiments

### 4.1. Experimental Setup

**Datasets.** To evaluate the generalizability of FLAG across varying tissue architectures, we utilize three representative cohorts from the HEST-1k benchmark (Jaume et al., 2024): HER2ST, KIDNEY, and PRAD. We adopt a random split of 7:2:1 for training, validation, and testing at the slide level to ensure robust evaluation. We implement a data-driven High-Mean & High-Variance Gene (HMHVG) selection strategy to identify biologically active targets from the training data.

**Baselines.** We benchmark against five state-of-the-art methods categorized into two paradigms. Discriminative Models: **HisToGene** (Pang et al., 2021), **BLEEP** (Xie et al.,

2023) and **TRIPLEX** (Chung et al., 2024). Generative Models: **Stem** (Zhu et al., 2025b) and **STFlow** (Huang et al., 2025). This selection covers both deterministic and generative approaches, providing a comprehensive performance landscape.

**Metrics.** We employ a dual-evaluation protocol. Gene-wise Accuracy: Pearson Correlation Coefficient (PCC) and Mean Squared Error (MSE) to assess numerical precision; Structural Fidelity: Gene Structural Correlation (GSC) and Spatial Structural Correlation (SSC) to assess the preservation of gene-gene and gene-spatial structures, respectively.

All experiments were conducted on one NVIDIA H800 GPU. Detailed experimental settings are listed in Appendix C.

### 4.2. Main Results

Table 1 presents the quantitative evaluation across three diverse dataset. Beyond standard performance metrics, the results highlight distinct behavioral differences between discriminative and generative paradigms, positioning FLAG as a unified solution that harmonizes gene-wise accuracy with structural fidelity.

discriminative methods have traditionally served as strong baselines for pointwise metrics (PCC, MSE) by directly minimizing reconstruction error. As shown in the Table 1, FLAG demonstrates highly competitive performance compared to these benchmark models, and in some cases even outperforms them. This indicates that incorporating generative prior does not diminish the model's ability to accurately recover individual gene expression levels. FLAG effectively maintains the high fidelity required for downstream analysis while offering additional structural benefits.

A key observation is that generative approaches (including Stem, STFlow, and FLAG) generally outperform discriminative methods in preserving GSC. This aligns with the expectation that diffusion and flow matching models are better suited to approximate the joint data distribution rather than treating genes as independent regression targets. By leveraging foundation model priors, FLAG further amplifies this advantage, demonstrating robust recovery of regulatory networks and co-expression manifolds.

The SSC metric reveals a key phenomenon in how different models handle spatial structure: While generative baseline models avoid the over-smoothing artifacts common in regression models, their lower SSC scores indicate insufficient alignment with the structural features of the real data. This suggests that these generative models may produce spatial variations that do not correctly correspond to the actual morphology of tissue. In contrast, FLAG achieves superior SSC scores across all datasets.

### 4.3. Analysis of Gene Dimension Curse

A fundamental challenge in applying diffusion models to spatial transcriptomics is the curse of dimensionality. As the number of target genes increases, the joint probability space expands exponentially, often leading to variance explosion and training instability. To investigate this, we evaluated model performance on the HER2ST dataset across varying gene panel sizes $G \in \{10, 50, 100, 200, 400, 800\}$.

As illustrated in Figure 4, the Joint Node-Edge Diffusion (red line), which attempts to denoise both node features and graph structures simultaneously, suffers from catastrophic collapse. Its performance peaks at low dimensions ($G = 10$) but degrades rapidly as $G$ increases, dropping to near-zero correlation at $G = 800$. The Node-Only Diffusion (blue line) mitigates this collapse to some extent by fixing the graph topology and performing denoising only on node features. However, it still exhibits a clear downward trend, indicating that purely geometric constraints are insufficient to handle the semantic complexity of hundreds of genes.

In contrast, FLAG (green line) maintains high fidelity even in high-dimensional settings. Notably, at $G = 800$, FLAG retains a PCC significantly superior to the Node-Only diffusion / Joint Node-Edge diffusion. This result confirms that FLAG can effectively overcome dimensionality constraints, enabling its scalability to larger-scale and biologically more comprehensive gene panels.

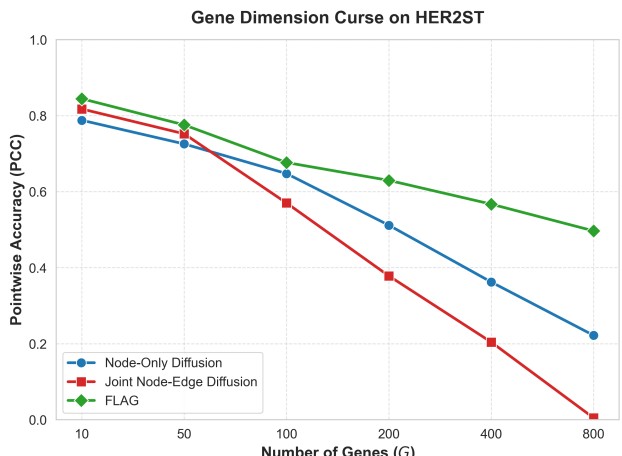

*Figure 4.* **Gene Dimension Curse on HER2ST.** We compare the PCC of FLAG against baseline diffusion strategies across varying gene panel sizes ($G$).

### 4.4. Ablation Study

To validate FLAG's design and explicitly attribute the performance gains, we ablated key components on the HER2ST dataset (Table 2). Crucially, to address whether the improvements stem solely from the spatial graph prior, we replaced the diffusion backbone with a deterministic super-

vised regressor equipped with the exact same spatial graph ("w/o Diffusion"). While this supervised variant maintains a reasonable pointwise accuracy (PCC 0.6748), its structural fidelity severely collapses, with GSC dropping to 0.3217 and SSC to 0.5685. This empirically proves that the generative diffusion engine is indispensable for preventing oversmoothing.

Furthermore, removing the GFM Alignment ("w/o GFM Alignment") noticeably degrades predictive accuracy and spatial coherence, confirming the necessity of biological priors. Conversely, removing the spatial encoder ("w/o Spatial Graph") severely compromises structural fidelity (SSC). In summary, FLAG's components are functionally synergistic: the spatial graph provides the structural canvas, GFM alignment enforces biological rules, and the diffusion framework acts as the generative engine. Appendix H further investigates GFM architectural variants and provides extensive sensitivity analyses on the spatial graph construction.

*Table 2.* **Ablation study on HER2ST.** We verify the necessity of key components. "w/o Diffusion (Supervised)" replaces the generative backbone with a deterministic regressor while keeping the spatial graph. "w/o GFM Alignment" removes the Foundation Gene Alignment, and "w/o Spatial Graph" removes the spatial backbone. FLAG integrates all to achieve optimal performance.

| Method Variant | PCC ↑ | MSE ↓ | GSC ↑ | SSC ↑ |
|---|---|---|---|---|
| w/o Diffusion (Supervised) | 0.6748 | 0.7864 | 0.3217 | 0.5685 |
| w/o GFM Alignment | 0.6682 | 0.7938 | 0.8713 | 0.5894 |
| w/o Spatial Graph | 0.6297 | 0.8499 | **0.9028** | 0.3399 |
| **FLAG** | **0.6835** | **0.7342** | 0.8926 | **0.6386** |

## 4.5. Downstream Evaluations

The ultimate goal of predicting spatial transcriptomics is enabling reliable biological discovery, rather than merely pointwise numerical fitting. To demonstrate that the structural fidelity captured by GSC and SSC directly translates to clinical utility, we evaluate FLAG on downstream tasks using the HER2ST dataset.

**Gene-Level Structure and DEG Identification.** We first examine the *Estrogen Response Early* pathway (from MSigDB (Liberzon et al., 2015)), a densely co-regulated module critical in breast cancer pathology. In Figure 5, the Ground Truth exhibits distinct block-diagonal functional cliques. FLAG effectively restores these crisp regulatory boundaries by leveraging its GFM alignment prior. Generative baselines like Stem capture only the coarse layout, while discriminative methods (e.g., HisToGene) severely collapse the expression variance, producing unstructured, over-smoothed matrices. Crucially, preserving these internal regulatory networks dramatically improves Differentially Expressed Gene (DEG) discovery. Extracted using ground-truth spatial domains as masks, FLAG consistently achieves the highest overlap with true marker genes (Table 3, e.g., 0.5

at Top-50). This confirms that FLAG reliably recovers functional biomarkers rather than simply hallucinating plausible expressions.

**Spatial Texture and Domain Identification.** To evaluate the preservation of tissue-level organization (SSC), Figure 6 visualizes spatial autocorrelation via the Moran's I scatter plot. Ideally, points should align tightly along the diagonal ($y = x$). FLAG closely approximates this ideal state, preserving the authentic morphological heterogeneity of the tissue. In contrast, STFlow tends to lie above the diagonal: its graph attention backbone tends to over-smooth spot-spot interactions, thereby producing "hyper-correlated" mappings that exaggerate spatial continuity and flatten fine-grained heterogeneity. The Stem model exhibits the opposite behavior, with points predominantly distributed along the diagonal below: due to the absence of a clear spatial pattern, this model underestimates autocorrelation and generates fragmented spatial patterns. Discriminative baselines deviate the most from the diagonal, reflecting their limitations in reconstructing spatial structures at the distribution level rather than merely their accuracy at the gene level. In the Fig. 7a, we visualized the spatial expression map of the marker gene *ERBB2*, FLAG achieves a better match with the ground truth. For the downstream task of spatial clustering (Fig. 7b), FLAG recovers distinct tissue categories with clear boundaries. We also quantify this clustering performance on the HER2ST slide (SPA148). As reported in Table 3, FLAG outperforms all baselines by a large margin. More experiments are shown in Appendix H.

*Table 3.* **Quantitative Downstream Evaluations on HER2ST.** We evaluate DEG Identification consistency (Top-20 and Top-50 overlap ratio ↑) and Spatial Domain Identification (ARI ↑, NMI ↑) against ground truth.

| Method | DEG Overlap Ratio | | Spatial Domain Clustering | |
|---|---|---|---|---|
| | Top-20 ↑ | Top-50 ↑ | ARI ↑ | NMI ↑ |
| HisToGene | 0.1333 | 0.3644 | 0.4733 | 0.6974 |
| BLEEP | 0.2188 | 0.3800 | 0.5984 | 0.7265 |
| TRIPLEX | 0.1571 | 0.2829 | 0.6744 | 0.7867 |
| Stem | 0.3286 | 0.4686 | 0.5303 | 0.7139 |
| STFlow | 0.2857 | 0.4133 | 0.5998 | 0.7754 |
| **FLAG** | **0.3944** | **0.5000** | **0.8451** | **0.9140** |

## 5. Conclusion

In this paper, we introduced FLAG. By harmonizing semantic guidance from pretrained foundation models with a spatially-aware graph condition, FLAG propels the recovery of intricate spatial structures to a new level of fidelity while effectively maintaining state-of-the-art pointwise prediction accuracy. Our extensive experiments on several datasets demonstrate that FLAG successfully bridges the gap between precise gene expression estimation and biologically distribution consistent. We believe that FLAG paves the

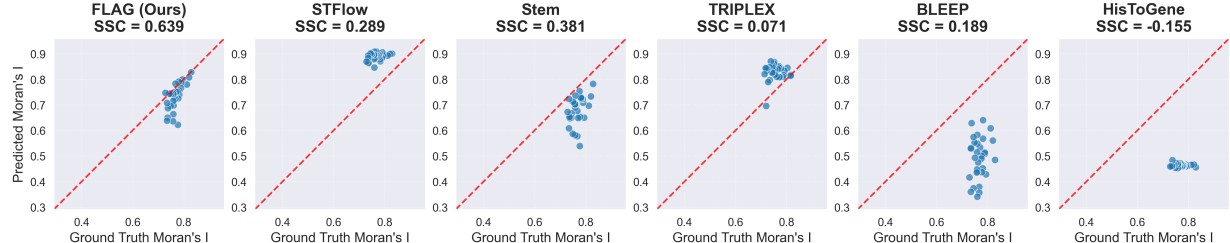

*Figure 5.* **Recovery of Gene Regulatory Networks** The co-expression matrices for genes in the intersection of the *Estrogen Response Early* pathway and the top-200 HMHVG panel.

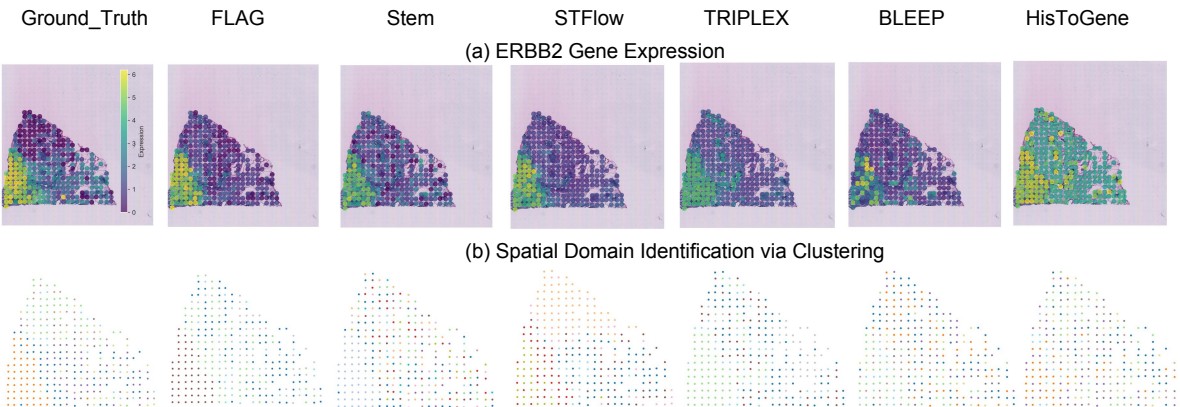

*Figure 6.* **GT vs Predicts Moran's I scatter.** The predicted Moran's I against the Ground Truth for top-32 spatially variable genes.

*Figure 7.* **Evaluation on spatial relationships. (a) Single-gene Spatial Pattern Recovery.** The spatial expression map of the representative marker gene *ERBB2*. **(b) Spatial Domain Identification via Clustering.** An unsupervised method to cluster spatial gene expression.

way for scalable, biologically consistent spatial transcriptomics generation, offering a powerful tool for computational pathology.

In the future, several directions are worth exploring. First, while FLAG performs well, the iterative nature of diffusion models implies a trade-off between speed and quality. Future work could investigate acceleration methods to reduce inference latency. Second, biological tissues are inherently three-dimensional. Extending our graph backbone to capture 3D volumetric dependencies from serial sections remains an exciting avenue for future research. Finally, our current validation focuses on within-tissue cohorts, leaving zero-shot cross-tissue generalization as an important open challenge for future work.

## Acknowledgments

This work was supported by the National Natural Science Foundation of China (Nos. 82394432, 92249302), the Shanghai Municipal Science and Technology Major Project (No. 2023SHZDZX02), and the AI for Science Program of the Shanghai Municipal Commission of Economy and Information. We also acknowledge the support of the Novalnspire platform (Shanghai Academy of Artificial Intelligence for Science) and the CFFF platform (Fudan University) for providing computational resources.

## Impact Statement

This paper presents work whose goal is to advance the field of Machine Learning for spatial transcriptomics gene prediction. There are many potential societal consequences of our work, none which we feel should be highlighted here.

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

# A. Additional Related Work

**Discriminative Models for ST Predictions**   Early studies framed WSI-to-ST prediction as supervised regression from histology to spot-level gene expression. ST-Net (He et al., 2020) and DeepSpaCE (Monjo et al., 2022) employed CNN encoders to map patch features to gene expression independently for each spot. Meanwhile, some works expanded the architectural capacity using Transformers, such as HisToGene (Pang et al., 2021) applied a Vision Transformer to model long-range contextual relationships among image patches. A complementary line of work incorporated multi-resolution context, THItoGene (Jia et al., 2024) introduced dynamic convolution and capsule networks to detect subtle morphological cues associated with transcriptomic variation. TRIPLEX (Chung et al., 2024) integrating spot-level, neighborhood-level, and global whole-slide features via dedicated multi-scale encoders and fusion modules to improve prediction accuracy, and MERGE (Ganguly et al., 2025) refined multi-resolution fusion with interpretable attention mechanisms. Additional efforts explored representation alignment and generalization, including BLEEP (Xie et al., 2023) which used cross-modal contrastive learning to couple morphology and expression, Img2ST-Net (Zhu et al., 2025a) cast the task as image-to-image translation, and DANet (Wu et al., 2025) which applied domain adaptation for cross-cohort robustness.

**Graph-based Spatial Modeling**   Early methods such as SpaGCN (Hu et al., 2021) integrated expression, location, and histology to identify spatial domains, and Squidpy (Palla et al., 2022) established a general graph-based analysis framework that enables neighborhood enrichment and spatial connectivity modeling. STAligner (Zhou et al., 2023) further leveraged graph message passing for multi-sample alignment, while GraphST (Long et al., 2023) showed that spatial graphs capture meaningful microenvironmental structure across imaging-based and transcriptomic datasets. More recently, SEPAL (Mejia et al., 2023) and EGGN (Yang et al., 2024) introduced graph networks into WSI-to-ST prediction by propagating contextual information across histology-derived neighborhood graphs. These approaches highlight the importance of spatial topology but rely on deterministic message passing, limiting their ability to model uncertainty in high-dimensional gene features.

**Graph Diffusion**   DiGress (Vignac et al., 2023) proposed discrete denoising diffusion for categorical graph structures, and the recent Graph Diffusion Transformer (Liu et al., 2024) extended diffusion to multi-conditional molecular graphs with improved scalability and expressiveness.

# B. Method preliminaries

**Score-based Diffusion**   We adopt an SDE-based score model (Song et al., 2021), specifically a variance-exploding (VE) SDE (Karras et al., 2022) with a log-linear noise schedule $\sigma(t)$ over $t \in [0, 1]$. Let $x \in \mathbb{R}^G$ denote the spot-level gene expression vector, the forward noising process admits a closed-form perturbation

$$x_t = x_0 + \sigma(t)\,z, \qquad z \sim \mathcal{N}(0, I), \tag{10}$$

and corresponds to the Itô SDE

$$\mathrm{d}x_t = g(t)\,\mathrm{d}w_t, \qquad g(t) = \sqrt{\frac{\mathrm{d}}{\mathrm{d}t}\,\sigma(t)^2}, \tag{11}$$

where $w_t$ is a standard Wiener process and $x_0 \sim p_{\text{data}}$. Score-based models learn a time-dependent score function $s_\theta(x, t, c)$ conditioned on auxiliary information $c$. For the VE-SDE, denoising score matching reduces to the supervised objective:

$$\mathcal{L}_{\text{score}} = \mathbb{E}_{t \sim \mathcal{U}(0,1)}\mathbb{E}_{x_0 \sim p_{\text{data}},\, z \sim \mathcal{N}(0, I)} \\ \|s_\theta(x_t, t, c) - \nabla_{x_t} \log p_t(x_t \mid x_0)\|_2^2. \tag{12}$$

Let $p_t(x)$ denote the marginal distribution of $x_t$ under the forward SDE. Once $s_\theta$ is learned, sampling can proceed via the reverse-time SDE or via the associated probability flow ODE (PF-ODE) (Song et al., 2021):

$$\frac{\mathrm{d}x_t}{\mathrm{d}t} = -\tfrac{1}{2}g(t)^2\,s_\theta(x_t, t, c), \tag{13}$$

which deterministically transports samples from the prior distribution at $t = 1$ (a broad Gaussian with variance $\sigma_{\max}^2$) to the data distribution at $t = 0$. In practice, we integrate (13) numerically with a second-order Heun / RK2 solver (Karras et al., 2022), starting from $x_{t=1} \sim \mathcal{N}(0, \sigma_{\max}^2 I)$ under the learned score field $s_\theta$.

**Graph Diffusion** In contrast to Euclidean diffusion where only a continuous vector $x_t$ is corrupted and denoised over time, graph diffusion additionally perturbs the relational structure between samples. Let $X \in \mathbb{R}^{S \times G}$ denote the spot gene expression as node attribute, and let $A \in \mathbb{R}^{S \times S}$ denote the edge attributes encoding spot–spot relationship (e.g., spatial adjacency, gene correlation). Following prior joint node–edge diffusion works (Vignac et al., 2023), we define a forward stochastic process that simultaneously corrupts node and edge states:

$$\mathrm{d}X_t = g_X(t)\,\mathrm{d}w_t^X, \qquad \mathrm{d}A_t = g_A(t)\,\mathrm{d}w_t^A, \tag{14}$$

where $w_t^X$ and $w_t^A$ are independent Wiener processes. To learn the joint score field, we optimize a weighted sum of denoising score matching objectives for both nodes and edges:

$$\mathcal{L}_{\mathrm{graph}} = \mathbb{E}_{t \sim \mathcal{U}(0,1)} \mathbb{E}_{(X_0, A_0),\, z_X, z_A}$$
$$\left[ \left\| s_\theta^{(X)}(X_t, A_t, t, c) - \nabla_{X_t} \log p_t(X_t \mid X_0) \right\|_2^2 \right. \tag{15}$$
$$\left. + \left\| s_\theta^{(A)}(X_t, A_t, t, c) - \nabla_{A_t} \log p_t(A_t \mid A_0) \right\|_2^2 \right].$$

This extends score-based diffusion by treating the entire spatial expression field $(X, A)$ as the evolving state. The reverse dynamics admit an associated PF-ODE, which provides a deterministic sampler analogous to the Euclidean case:

$$\frac{\mathrm{d}X_t}{\mathrm{d}t} = -\tfrac{1}{2} g_X(t)^2\, s_\theta^{(X)}(X_t, A_t, t, c),$$
$$\frac{\mathrm{d}A_t}{\mathrm{d}t} = -\tfrac{1}{2} g_A(t)^2\, s_\theta^{(A)}(X_t, A_t, t, c). \tag{16}$$

where $s_\theta^{(X)}$ and $s_\theta^{(A)}$ denote the node and edge components of the joint score. This formulation couples all spots through the evolving edge structure, and can improve spatially coordinated denoising ability over treating spots independently.

## C. Experimental Details

### C.1. Data Preprocessing and Gene Selection

**HMHVG Selection strategy** We select prediction targets using a statistical intersection strategy. The process ensures targets are both highly expressed and variable:

1. **Filtering**: We strictly use only the *training* slides to calculate statistics. Common genes present across all slides are identified first.

2. **Ranking**: For each gene $g$, we compute the mean expression $\mu_g$ and standard deviation $\sigma_g$ across all training spots.

3. **Intersection**: We rank genes by $\mu_g$ and $\sigma_g$ in descending order. The final panel $\mathcal{S}$ is the intersection of the top-$K_{search}$ genes from both lists:

$$\mathcal{S} = \{g \mid \mathrm{Rank}(\mu_g) \leq K_{search}\} \cap \{g \mid \mathrm{Rank}(\sigma_g) \leq K_{search}\}. \tag{17}$$

$K_{search}$ is adjusted dynamically to yield exact target sizes of $G \in \{50, 100, 200, 400, 800\}$ for the gene dimensional analysis. For all standard benchmarking experiments, we fix $G = 200$.

**Dataset Specifications** Table 4 details the statistics of the three cohorts used.

- **Preprocessing:** Raw gene counts are log1p-normalized: $x = \log(1 + x_{raw})$.

- **Patching:** WSIs are tiled into $224 \times 224$ patches centered on each spot.

*Table 4.* Detailed statistics of the datasets (HEST-1k subset). Split ratio: 7:2:1 (Slide-level).

| Dataset | Total Slides | Total Spots | Mean Spot/Slide |
|---------|-------------|-------------|-----------------|
| HER2ST | 36 | 13620 | 378 |
| KIDNEY | 23 | 25944 | 1128 |
| PRAD | 23 | 62710 | 2726 |

## C.2. Structural Metrics Definitions and Hyperparameter Rationale

**Gene Structural Correlation (GSC)**  This metric evaluates whether the prediction preserves the intrinsic gene-gene interactions.

- **Calculation:** Let $\mathbf{X} \in \mathbb{R}^{N \times G}$ be the expression matrix for a gene set. We first compute the **Gene-Gene Correlation Matrix** $\mathbf{C} \in \mathbb{R}^{G \times G}$. Standardization is applied first: $\tilde{\mathbf{X}} = (\mathbf{X} - \mu)/(\sigma + \epsilon)$. The correlation matrix is then computed as:

$$\mathbf{C} = \frac{1}{N-1}\tilde{\mathbf{X}}^{\top}\tilde{\mathbf{X}}. \tag{18}$$

- **Definition:** Let $\mathbf{v}_{GT}$ and $\mathbf{v}_{Pred}$ be the vectorized upper-triangular elements (excluding the diagonal) of the ground truth and predicted correlation matrices, respectively. GSC is defined as their Pearson correlation:

$$\text{GSC} = \text{Corr}(\mathbf{v}_{GT}, \mathbf{v}_{Pred}). \tag{19}$$

A high GSC indicates that the global topology of the gene regulatory network is preserved.

**Spatial Structural Correlation (SSC)**  This metric evaluates whether the model preserves the spatial texture intensities. To address potential concerns regarding hyperparameter selection for spatial evaluation, we clarify the rationale behind our choices.

- **Spatial Weight Matrix (W):** We construct a symmetric $k$-nearest neighbor graph based on spatial coordinates to define connectivity. Let $\mathcal{N}_k(i)$ be the set of $k$ nearest neighbors for spot $i$. The adjacency matrix $\mathbf{W} \in \{0,1\}^{N \times N}$ is defined as:

$$W_{ij} = 1 \quad \text{if } j \in \mathcal{N}_k(i) \text{ or } i \in \mathcal{N}_k(j), \quad \text{else } 0. \tag{20}$$

**Rationale for $k = 8$:** We set $k = 8$ to capture local microenvironments. Biologically, in typical spatial transcriptomics arrays, a central spot is immediately surrounded by 8 physical neighbors on the grid. Setting $k = 8$ robustly encapsulates this immediate cellular neighborhood without incorporating distant spots.

- **Rationale for Patch Size** ($224 \times 224$)**:** We crop H&E images into $224 \times 224$ patches to align with the default input specifications of external pathology foundation models (e.g., UNI) used for feature extraction. This standardization ensures architectural compatibility and provides a unified feature space for calculating patch-to-patch distances during spatial metric evaluation.

- **Moran's I Calculation:** For a gene $g$, let $\mathbf{x}_g \in \mathbb{R}^N$ be its expression vector and $\mathbf{z}_g = \mathbf{x}_g - \bar{x}_g$ be the centered vector. The global Moran's I is computed in matrix form:

$$I(g) = \frac{N}{S_0}\frac{\mathbf{z}_g^{\top}\mathbf{W}\mathbf{z}_g}{\mathbf{z}_g^{\top}\mathbf{z}_g}, \quad \text{where } S_0 = \sum_{i,j} W_{ij}. \tag{21}$$

- **Definition:** For each gene in HMHVG, we compute Moran's I, from the ground-truth and predicted expression maps, obtaining vectors $\mathbf{I}_{GT}, \mathbf{I}_{Pred}$. SSC is then defined as the Pearson correlation between these two vectors:

$$\text{SSC} = \text{Corr}(\mathbf{I}_{GT}, \mathbf{I}_{Pred}). \tag{22}$$

A high SSC indicates that the model preserves the global pattern of spatial autocorrelation across genes.

# D. Graph Diffusion Details

In this section, we provide the detailed engineering specifications of the model used to parameterize the joint score function $\mathbf{s}_\theta(\mathbf{X}_t, \mathbf{A}_t, t, \mathcal{C})$.

## D.1. Input Data Structures and Dimensions

The model processes a batch of spatial graphs. Let $B$ denote the batch size, $N$ the number of spots in a patch (variable), and $G$ the gene panel size. The input tensors are defined as follows:

- **Noisy Diffusion Inputs:**
    - **Node Features ($\mathbf{X}_t$):** Shape $(B, N, G)$. Represents the noisy gene expression state.
    - **Edge Features ($\mathbf{A}_t$):** Shape $(B, N, N, 1)$. Represents the noisy correlation matrix.

- **Conditioning Inputs:**
    - **Node Condition ($\mathbf{C}_v$):** Shape $(B, N, 1024)$. Visual features extracted via the UNI encoder.
    - **Edge Condition ($\mathbf{C}_e$):** Shape $(B, N, N, 2)$. A composite tensor where channel 0 contains physical proximity weights (Gaussian RBF) and channel 1 contains visual cosine similarity.
    - **Timestep ($t$):** A scalar $t \in [0, 1]$ sampled from $\mathcal{U}(0, 1)$.

## D.2. Graph Transformer Block Details

Each block $l$ transforms node features $\mathbf{H}_x^{(l)} \in \mathbb{R}^{N \times D}$ and edge features $\mathbf{H}_e^{(l)} \in \mathbb{R}^{N \times N \times D}$ into updated states $(\mathbf{H}_x^{(l+1)}, \mathbf{H}_e^{(l+1)})$. The computation proceeds in three stages:

**1. Adaptive Layer Normalization (AdaLN)** First, we fuse the time embedding $t_{emb}$ with pooled representations of the conditions to form a global context vector $z$. This vector regresses the scale and shift parameters for normalization:

$$z = \text{MLP}_{\text{fuse}}([t_{emb}, \text{Pool}(\mathbf{C}_v), \text{Pool}(\mathbf{C}_e)]) \tag{23}$$

$$\hat{\mathbf{H}}_x = \text{AdaLN}(\mathbf{H}_x, z) = (1 + \gamma_x(z)) \odot \text{LN}(\mathbf{H}_x) + \beta_x(z) \tag{24}$$

$$\hat{\mathbf{H}}_e = \text{AdaLN}(\mathbf{H}_e, z) = (1 + \gamma_e(z)) \odot \text{LN}(\mathbf{H}_e) + \beta_e(z) \tag{25}$$

**2. Joint Structure Learning (Edge-Modulated Attention)** We compute the attention scores $\mathbf{S} \in \mathbb{R}^{N \times N \times H}$ by interacting node queries/keys with edge-based gating. Let $\mathbf{Q}, \mathbf{K}, \mathbf{V}$ be projections of $\hat{\mathbf{H}}_x$. The attention topology is computed as:

$$\mathbf{S}_{ij} = \left( \frac{\mathbf{q}_i \mathbf{k}_j^T}{\sqrt{d}} \right) \odot \underbrace{\left( 1 + \text{Linear}(\hat{\mathbf{H}}_{e,ij}) + \alpha \text{Linear}(\mathbf{C}_{e,ij}) \right)}_{\text{Structural Gating}} + \underbrace{\text{Linear}(\hat{\mathbf{H}}_{e,ij}) + \gamma \text{Linear}(\mathbf{C}_{e,ij})}_{\text{Structural Bias}} \tag{26}$$

where $\alpha, \gamma$ are learnable scalars initialized to 0.1. This matrix $\mathbf{S}$ captures the learned structural dependencies and serves as the source for updating both streams.

**3. Dual-Stream Updates** The structural information $\mathbf{S}$ is bifurcated to update nodes and edges:

- **Node Update.** The structural attention matrix $\mathbf{S}$ acts as standard attention weights to aggregate value vectors $\mathbf{V}$:

$$\mathbf{H}_x^{\text{attn}} = \mathbf{H}_x^{(l)} + \text{Lin}_{\text{out}}\big(\text{Softmax}(\mathbf{S}) \cdot \mathbf{V}\big). \tag{27}$$

- **Edge Update.** The raw score matrix $\mathbf{S}$ is also directly projected to update the edge features, ensuring that edge representations evolve consistently with the attention topology:

$$\mathbf{H}_e^{\text{attn}} = \mathbf{H}_e^{(l)} + \text{Lin}_{\text{edge}}(\mathbf{S}). \tag{28}$$

**4. Gated Feed-Forward Networks** Finally, both streams undergo point-wise processing via Gated-GELU networks. Unlike standard FFNs, GEGLU projects inputs into a gating stream and a value stream:

$$\text{FFN}(\mathbf{h}) = \mathbf{W}_2 \cdot (\text{GELU}(\mathbf{W}_{gate}\mathbf{h}) \odot (\mathbf{W}_{val}\mathbf{h})) \tag{29}$$

The block output is then computed with residual connections:

$$\mathbf{H}_x^{(l+1)} = \mathbf{H}_x^{attn} + \text{FFN}_{node}(\text{AdaLN}(\mathbf{H}_x^{attn}, z)) \tag{30}$$

$$\mathbf{H}_e^{(l+1)} = \mathbf{H}_e^{attn} + \text{FFN}_{edge}(\text{AdaLN}(\mathbf{H}_e^{attn}, z)) \tag{31}$$

### D.3. Consistency Loss Details

The structure-consistency loss $\mathcal{L}_{cons}$ bridges the node and edge diffusion streams. It is computed during training as follows:

1. **Denoise (Tweedie's Formula):** We estimate the clean data $\hat{\mathbf{X}}_0$ and $\hat{\mathbf{A}}_0$ from the current noisy states and predicted scores:

$$\hat{\mathbf{X}}_0 = \mathbf{X}_t + \sigma(t)^2 \mathbf{s}_\theta^X, \quad \hat{\mathbf{A}}_0 = \mathbf{A}_t + \sigma(t)^2 \mathbf{s}_\theta^A \tag{32}$$

2. **Empirical Correlation:** We compute the PCC of the estimated node expression within the batch:

$$\mathbf{P}_{pred} = \text{Corr}(\hat{\mathbf{X}}_0) = \frac{(\hat{\mathbf{X}}_0 - \mu)(\hat{\mathbf{X}}_0 - \mu)^T}{\sigma_x \sigma_x^T + \epsilon} \tag{33}$$

3. **Loss Computation:** We minimize the L1 distance between the explicitly predicted edge $\hat{\mathbf{A}}_0$ and the implicit node correlation $\mathbf{P}_{pred}$, masking out diagonal self-loops:

$$\mathcal{L}_{cons} = \frac{1}{N(N-1)} \sum_{i \neq j} \left| \hat{\mathbf{A}}_{0,ij} - \mathbf{P}_{pred,ij} \right| \tag{34}$$

## E. Analysis of the Gene Dimension Curse

We provide a simplified analysis to explain why jointly denoising node expressions and functional edges becomes harder as the gene dimension $G$ grows.

### E.1. Setup

Consider a WSI with $N$ spots. For each gene $g \in \{1, \ldots, G\}$, let $y_g \in \mathbb{R}^N$ denote its expression vector across spots. We adopt a simplified but standard model:

$$y_g \overset{\text{i.i.d.}}{\sim} \mathcal{N}(0, A^*), \qquad A^* \in \mathbb{R}^{N \times N}. \tag{35}$$

Stacking all genes gives $X_0 = [y_1, \ldots, y_G] \in \mathbb{R}^{N \times G}$, and the spot–spot Gram estimator is

$$\widehat{A} = \frac{1}{G} X_0 X_0^\top = \frac{1}{G} \sum_{g=1}^G y_g y_g^\top. \tag{36}$$

Thus $\widehat{A}$ is the sample covariance of $\{y_g\}_{g=1}^G$ with sample size $G$.

**Lemma A (Covariance estimation rate)** (Rigollet & Hütter, 2023). Let $y_1, \ldots, y_G \overset{iid}{\sim} \mathcal{N}(0, A^*) \in \mathbb{R}^N$ and $\widehat{A} = \frac{1}{G} \sum_{g=1}^G y_g y_g^\top$. Under standard bounded-spectrum conditions on $A^*$, the mean-squared Frobenius error satisfies

$$\mathbb{E}[\|\widehat{A} - A^*\|_F^2] = \Theta\left(\frac{N^2}{G}\right). \tag{37}$$

## E.2. Manifold thickness shrinks as $G$ grows

We analyze how the empirical functional connectivity $\widehat{\mathbf{A}} = \frac{1}{G} X_0 X_0^\top \in \mathbb{R}^{N \times N}$ concentrates as the gene panel size $G$ increases. Applying Lemma A with dimension $N$ and sample size $G$ yields

$$\mathbb{E}\big[\|\widehat{\mathbf{A}} - \mathbf{A}^*\|_F^2\big] = \Omega\left(\frac{1}{G}\right), \tag{38}$$

where $N$ is fixed for local spatial patches.

Equation (38) implies that the typical deviation of $\widehat{\mathbf{A}}$ from $\mathbf{A}^*$ scales as $\|\widehat{\mathbf{A}} - \mathbf{A}^*\|_F = \Omega(G^{-1/2})$. Consequently, the admissible $(\mathbf{X}, \mathbf{A})$ pairs satisfying $\mathbf{A} = f(\mathbf{X})$ concentrate into an increasingly thin neighborhood of the consistency manifold as $G$ grows.

## E.3. Sharpness of the Conditional Edge Score

We now translate the thinning of $\mathcal{M}$ into the sharpness of the conditional edge score $\nabla_{A_t} \log p(A_t \mid X_t)$.

Under the Gaussian approximation used in diffusion models,

$$p(A_t \mid X_t) \approx \mathcal{N}\big(\mu_t(X_t),\ \sigma_A^2(t)\,\Sigma_A(G, N)\big), \tag{39}$$

where $\Sigma_A(G, N)$ denotes the covariance of the Gram fluctuations in Eq. (38). Let

$$\Sigma_A(G, N) = U \Lambda U^\top \tag{40}$$

be its eigendecomposition, with $\Lambda = \mathrm{diag}(\lambda_1, \ldots, \lambda_D)$ and $D = N(N+1)/2$ the dimension of the symmetric edge space.

Since $N$ (and hence $D$) is fixed for a given tissue patch and Eq. (38) shows that $\mathbb{E}\|\widehat{A} - A^*\|_F^2 = \Theta(1/G)$, the typical magnitude of each eigenvalue satisfies

$$\lambda_k\big(\Sigma_A(G, N)\big) = O\left(\frac{1}{G}\right), \qquad k = 1, \ldots, D, \tag{41}$$

and in particular there exists a constant $C > 0$ such that

$$\lambda_{\min}\big(\Sigma_A(G, N)\big) \ \leq \ \frac{C}{G}. \tag{42}$$

In the eigen-basis $z = U^\top \mathrm{vec}(A_t)$, the coordinates are independent Gaussians

$$p(z_k \mid X_t) \sim \mathcal{N}(\mu_{t,k},\ \lambda_k), \tag{43}$$

and the corresponding score components are

$$s_k^*(z) := \frac{\partial}{\partial z_k} \log p(z_k \mid X_t) = -\frac{z_k - \mu_{t,k}}{\lambda_k}. \tag{44}$$

Since $\mathrm{Var}(z_k - \mu_{t,k}) = \lambda_k$, we obtain

$$\mathbb{E}\big[(s_k^*)^2\big] = \frac{1}{\lambda_k}. \tag{45}$$

The Fisher information matrix of $p(A_t \mid X_t)$ in this basis is diagonal with entries $\mathbb{E}[(s_k^*)^2]$. Its largest eigenvalue therefore satisfies

$$\lambda_{\max}\big(\mathrm{Fisher}(p(A_t \mid X_t))\big) = \max_k \frac{1}{\lambda_k} = \frac{1}{\lambda_{\min}(\Sigma_A(G, N))} \ \geq \ c\,G, \tag{46}$$

for some constant $c > 0$ independent of $G$. Intuitively, as $G$ grows the covariance of the Gram estimator shrinks like $O(1/G)$ along at least one direction, and the conditional score curvature along that direction is amplified by the inverse factor $\Omega(G)$. Because the edge space dimension $D$ is fixed by $N$, this stiff direction dominates the Fisher information in the large-$G$ regime.

### E.4. Optimization Lower Bound Divergence

Combining the Fisher scaling in Eq. (46) with a simple capacity assumption on the edge-score network, we obtain a linear-in-$G$ lower bound on the best achievable edge loss.

Let $\mathcal{F}_A$ be a class of edge-score networks (e.g., fixed-depth neural nets) with uniformly bounded Lipschitz constants. For such a class, approximating a score function whose Fisher information has largest eigenvalue $\lambda_{\max}$ necessarily incurs a squared error at least proportional to $\lambda_{\max}$. Applied to $p(A_t \mid X_t)$, whose Fisher matrix scales as $\lambda_{\max}(\text{Fisher}(p(A_t \mid X_t))) = \Theta(G)$ by Eq. (46), this implies that there exists a constant $c_1 > 0$, independent of $G$, such that the optimal edge-score approximation error satisfies

$$\mathcal{L}^{\star}_{\text{edge}}(G) := \inf_{s_\theta^{(A)} \in \mathcal{F}_A} \mathbb{E}\big[\|s_\theta^{(A)} - s^*(A_t \mid X_t)\|_F^2\big] \geq c_1 G. \tag{47}$$

The optimal value of the joint training objective at gene dimension $G$ is

$$\mathcal{L}^{\star}_{\text{joint}}(G) := \inf_{\theta_X, \theta_A} \mathcal{L}_{\text{joint}}(\theta_X, \theta_A; G). \tag{48}$$

By non-negativity of the consistency term,

$$\mathcal{L}^{\star}_{\text{joint}}(G) \geq \inf_{\theta_X, \theta_A} \big\{\mathcal{L}_{\text{node}}(\theta_X) + \mathcal{L}_{\text{edge}}(\theta_A; G)\big\}. \tag{49}$$

Since $\theta_X$ and $\theta_A$ are independent parameters, the infimum separates:

$$\inf_{\theta_X, \theta_A} \big\{\mathcal{L}_{\text{node}}(\theta_X) + \mathcal{L}_{\text{edge}}(\theta_A; G)\big\} = \inf_{\theta_X} \mathcal{L}_{\text{node}}(\theta_X) + \inf_{\theta_A} \mathcal{L}_{\text{edge}}(\theta_A; G). \tag{50}$$

We denote the node-only optimum by $\mathcal{L}^{\star}_{\text{node}} := \inf_{\theta_X} \mathcal{L}_{\text{node}}(\theta_X)$. Combining Eqs. (47)–(50) yields

$$\mathcal{L}^{\star}_{\text{joint}}(G) \geq \mathcal{L}^{\star}_{\text{node}} + c_1 G, \tag{51}$$

i.e.,

$$\mathcal{L}^{\star}_{\text{joint}}(G) - \mathcal{L}^{\star}_{\text{node}} \geq \Omega(G), \tag{52}$$

which is the gene dimensional phenomenon reported in the main text.

## F. Extraction of Gene Foundation Model Embeddings

In this section, we detail the implementation of the Gene Foundation Model Embedding extraction. Unlike static gene embeddings (e.g., gene2vec (Du et al., 2019)), we leverage the inference capabilities of pretrained Gene Foundation Model (GFM) to extract contextualized embeddings. These embeddings capture the functional state of genes and cells within the specific transcriptomic context of each spatial spot. All GFM weights are frozen during training, and embeddings are pre-computed offline.

### F.1. Overview

We employ three distinct foundation models to capture multi-scale biological priors:

- **Geneformer (Theodoris et al., 2023)**: Provides gene-level embeddings based on rank-based attention.

- **scGPT (Cui et al., 2024)**: Provides gene-level embeddings using value-binned attention.

- **CellPLM (Wen et al., 2024)**: Provides a holistic spot-level (cell-level) embedding.

### F.2. Gene-Level Embeddings (Geneformer & scGPT)

F.2.1. GENEFORMER: RANK-BASED CONTEXTUALIZATION

**Preprocessing & Input Construction** We utilize the `Geneformer-V2-316M` checkpoint. Geneformer is pretrained on rank-value encoded sequences. For each spot $i$, we first normalize raw counts and rank genes by expression value in descending order. Non-expressed genes are excluded. The input sequence is defined as:

$$S_i = \{t_{\pi(1)}, t_{\pi(2)}, \ldots, t_{\pi(L_i)}\} \tag{53}$$

where $t$ is the token ID for Ensembl gene IDs, $\pi(k)$ denotes the index of the gene with the $k$-th highest expression, and $L_i$ is the sequence length (bounded by the model's context window).

**Extraction**   We perform a forward pass to obtain the hidden states $\mathbf{H}^{GFM} \in \mathbb{R}^{L_i \times D_{GFM}}$ from the last transformer layer ($D_{GFM} = 512$). A critical challenge is that the output sequence follows the expression rank order, which varies across spots. To align this with our fixed gene panel $\mathcal{S}$, we implement a reverse mapping strategy:

1. We initialize a zero-filled embedding matrix $\mathbf{E}_i \in \mathbb{R}^{|\mathcal{S}| \times D_{GFM}}$, where $|\mathcal{S}|$ denotes the number of genes in the target panel.

2. For each position $k$ in the output sequence, we decode the token $t_{\pi(k)}$ back to its Ensembl ID.

3. If this ID corresponds to the $j$-th gene in our target panel $\mathcal{S}$ (i.e., $g_j \in \mathcal{S}$), we populate the matrix: $\mathbf{E}_{i,j} \leftarrow \mathbf{H}_k^{GFM}$.

Genes not present in the top-ranked context of spot $i$ remain as zero vectors and are masked during the alignment loss calculation.

### F.2.2. SCGPT: VALUE-BINNED CONTEXTUALIZATION

**Preprocessing & Input Construction**   We utilize the `scGPT-human` checkpoint. Unlike Geneformer, scGPT utilizes explicit value embeddings. We align the input tokens to our fixed gene panel $\mathcal{G}$. The expression values are discretized into bins $b(x_{i,j})$ consistent with the official implementation. The input sequence is constructed as:

$$S_i = \big\{ \texttt{[CLS]}, \ g_1 \oplus b(x_{i,1}), \ g_2 \oplus b(x_{i,2}), \dots, \ g_M \oplus b(x_{i,M}) \big\} \tag{54}$$

where $\oplus$ denotes the element-wise addition of gene identity and expression value embeddings.

**Extraction**   We extract the contextualized representations from the last transformer block. Since the input strictly follows the order of $\mathcal{G}$, no reverse mapping is required. We discard the special `[CLS]` token at index 0 and slice the output tensor to obtain the gene embeddings:

$$\mathbf{E}_i^{sc} = \mathbf{H}_{i,1:M+1,:}^{sc} \in \mathbb{R}^{M \times D_{sc}} \tag{55}$$

where $D_{sc}$ is the hidden dimension of scGPT.

### F.3. Spot-Level Embeddings (CellPLM)

**Extraction**   We utilize the `CellPLM-85M` checkpoint, which is optimized for encoding cell-type and cell-state semantics. For each spot $i$, the selected gene expression profile is fed into the model pipeline. We extract the projection output (latent representation) denoted as $\mathbf{e}_i^{spot} \in \mathbb{R}^{D_{cp}}$, where $D_{cp} = 512$.

**Gene-wise Pooling for Alignment**   Since FLAG operates on gene-wise tokens $\mathbf{H}^{(K)} \in \mathbb{R}^{|\mathcal{S}| \times D_{GFM}}$, we cannot directly align them to the single vector $\mathbf{e}_i^{spot}$. Instead, we aggregate the semantic information of FLAG's output via a mean pooling operation:

$$\mathbf{h}_i^{pool} = \frac{1}{|\mathcal{S}|} \sum_{j=1}^{|\mathcal{S}|} \mathbf{H}_{i,j,:}^{(K)} \tag{56}$$

The alignment loss for the spot-level prior is then defined as:

$$\mathcal{L}_{align} = \frac{1}{N} \sum_{i=1}^{N} \| \mathcal{P}_{spot}(\mathbf{h}_i^{pool}) - \mathbf{e}_i^{spot} \|_2^2 \tag{57}$$

where $\mathcal{P}_{spot}$ is a learnable projector mapping the pooled FLAG representation to the CellPLM embedding space.

## F.4. Other Implementation Details

All preprocessing steps were performed offline to ensure efficient training.

- **Gene Mapping:** We mapped HGNC symbols (Seal et al., 2023) to Ensembl IDs using MyGeneInfo (Xin et al., 2016) to ensure compatibility with Geneformer and scGPT vocabularies.

- **Filtering:** Spots with zero expression for all selected genes were excluded from the embedding extraction process to maintain numerical stability.

# G. FLAG Implementation Details

This part provides the precise engineering specifications for FLAG, and we focus on the unique mechanism and the specialized solver implementation.

## G.1. Construction of Observable Topology

We pre-compute a fixed tissue graph that encodes spatial structure. For each spot $i$, let $\mathbf{u}_i \in \mathbb{R}^2$ denote its spatial coordinates on the slide (spot center in pixel), and let $\mathbf{v}_i \in \mathbb{R}^{d_v}$ denote its visual feature extracted by the pretrained WSI encoder (e.g., UNI).

For every pair of spots $(i, j)$, we build a 2-dimensional edge attribute $\mathcal{C}_{e,ij} \in \mathbb{R}^2$ composed of a distance kernel and an image-similarity term:

$$w_{\text{dist}}(i, j) = \exp\left(-\frac{\|\mathbf{u}_i - \mathbf{u}_j\|_2^2}{2\sigma^2}\right),$$
$$w_{\text{img}}(i, j) = \text{CosSim}(\mathbf{v}_i, \mathbf{v}_j), \tag{58}$$

and define

$$\mathcal{C}_{e,ij} = \left[w_{\text{dist}}(i, j), \ w_{\text{img}}(i, j)\right]. \tag{59}$$

Here $\sigma$ is a length-scale hyperparameter controlling how quickly the distance kernel decays, and we set $\sigma$=224. CosSim denotes cosine similarity between visual embeddings. Throughout training, this observable topology $\mathcal{C}_e = \{\mathcal{C}_{e,ij}\}$ is kept *fixed* and only serves as an immutable structural prior for the graph encoder, thereby avoiding the high-variance edge generation that leads to the collapse analyzed in Appendix E.

## G.2. Architecture Details

To clarify the structural versatility of our model, we illustrate the two operating modes of the Spatial Graph Encoder in Figure 8. While the core architecture stacked Graph Transformer Blocks (GTBs) remains shared, the flow of edge information differs fundamentally between the pure Graph Diffusion (Joint Node-Edge Diffusion) task and the FLAG foundation model framework.

SPATIAL ENCODING IN FLAG (STATIC MODE)

In the FLAG framework, we utilize the encoder in **Mode 2**. Unlike the generative graph diffusion process where edges are actively denoised, FLAG treats the tissue topology as a fixed constraint. The encoder actively routes the noisy gene state $\mathbf{X}_t$ through this spatial backbone, where spatial mixing is governed by a specialized, statically-modulated attention operator.

Let $\mathbf{Q}, \mathbf{K}$ be projections of the noisy gene features $\mathbf{X}_t$. The attention scores $\mathbf{S} \in \mathbb{R}^{N \times N}$ are modulated solely by the fixed edge condition $\mathcal{C}_e$ (representing spatial proximity or visual adjacency) via learnable scalars $\alpha, \gamma$:

$$\mathbf{S}_{ij} = \left(\frac{\mathbf{q}_i \mathbf{k}_j^\top}{\sqrt{d}}\right) \odot (1 + \alpha \cdot \text{Linear}(\mathcal{C}_{e,ij})) + \gamma \cdot \text{Linear}(\mathcal{C}_{e,ij}). \tag{60}$$

Crucially, contrast this with Mode 1 (Figure 8, Left), where an additional dynamic edge term $\mathbf{E}_t$ would modulate the attention. In FLAG, we enforce a fixed-topology constraint by strictly setting the dynamic edge term to zero.

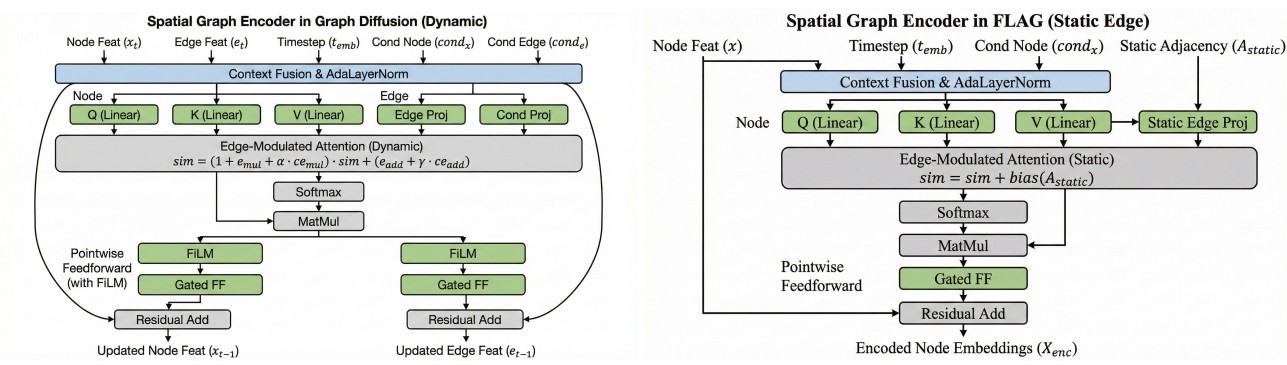

*Figure 8.* **Dual-Mode Spatial Graph Encoder. Left (Mode 1):** In the Graph Diffusion setting, the encoder operates dynamically, where both nodes $\mathbf{X}_t$ and edges $\mathbf{E}_t$ are noisy latent variables updated at each timestep. The attention mechanism is modulated by the evolving edge features. **Right (Mode 2):** In the FLAG framework, the encoder functions as a spatial feature extractor. The dynamic edge evolution is suppressed ($A_t \to 0$), and the topology is strictly defined by the fixed spatial adjacency $\mathcal{A}_{static}$ (or $\mathcal{C}_e$), ensuring the node representations are aligned with the immutable spatial context of the tissue.

The output of this static graph backbone, denoted as $\mathbf{H}_{\text{spatial}}$, encapsulates the spatially-contextualized gene features. This representation is then projected to the DiT's conditional space via a 3-layer bottleneck MLP:

$$\mathbf{C}_{\text{graph}} = \text{Linear}_{D_{\text{hid}}} \circ \text{SiLU} \circ \text{Linear}_{D_{\text{gene}}}(\mathbf{H}_{\text{spatial}}). \tag{61}$$

This $\mathbf{C}_{\text{graph}}$ is added to the timestep embedding to form the global condition for the gene DiT, guiding the generation process with spatially-aware priors.

### G.3. Algorithms

We present the pseudocode of the overall train and inference processes of FLAG.

---

**Algorithm 1** FLAG Training Procedure

---

1: **Input:** Clean genes $\mathbf{X}_0$, Conditions $\mathcal{C}_v, \mathcal{C}_c$, Frozen GFM embeddings $F$.
2: **Init:** Construct fixed edges $\mathcal{C}_e$ (Appendix E.1).
3: *// 1. Forward Diffusion*
4: Sample $t \sim \mathcal{U}(0, 1)$ and $\epsilon \sim \mathcal{N}(0, \mathbf{I})$.
5: $\mathbf{X}_t = \mathbf{X}_0 + \sigma(t)\epsilon$.
6: *// 2. Spatial Denoising*
7: **Pass:** $\mathrm{H}_{spatial} = \text{GraphBackbone}(\mathbf{X}_t, \mathcal{C}_e, t)$.
8: **Project:** $\mathcal{C}_{cond} = \text{MLP}(\mathrm{H}_{spatial}) + t_{emb}$.
9: *// 3. Gene Generation (DiT with intermediate feature)*
10: **Pass:** $(S_\theta, \mathrm{Z}_{inter}) = \text{DiT}(\mathbf{X}_t, \mathcal{C}_{cond}; \texttt{return\_layer} = N)$, *where $\mathrm{Z}_{inter}$ is the hidden state at the $N$-th Transformer block.*
11: *// 4. Optimization*
12: $\mathcal{L}_{diff} = \|\epsilon + S_\theta \cdot \sigma(t)\|^2$.
13: $\mathcal{L}_{align} = -\text{CosineSim}(\text{Normalize}(\mathrm{Z}_{inter}), \text{Normalize}(\mathbf{H}))$.
14: Update $\theta \leftarrow \theta - \nabla_\theta(\mathcal{L}_{diff} + \lambda\mathcal{L}_{align})$.

---

### G.4. Hyperparameter Configuration

Table 5 details the specific settings used for experiments.

---

**Algorithm 2** FLAG Inference Procedure

---

1: **Input:** Target $\mathcal{C}_v, \mathcal{C}_c$. Steps $K = 100$.
2: **Init:** $\mathbf{X} \sim \mathcal{N}(0, \sigma_{max}^2 \mathbf{I})$. Construct $\mathcal{C}_e$.
3: **for** $i = K - 1$ to $0$ **do**
4:     $t = t_i, t_{next} = t_{i-1}$.
5:     $\Delta t = t - t_{next}$.
6:     *// 1. Predictor Step (Euler)*
7:     $\mathcal{C}_{cond} = \text{MLP}(\text{GraphBackbone}(\mathbf{X_t}, \mathcal{C}_e, t)) + t_{i_{emb}}$.     ▷ *Recompute spatial context*
8:     $d_1 = -0.5 \cdot g(t)^2 \cdot \text{DiT}(\mathbf{X}, \mathcal{C}_{cond})$.
9:     $\mathbf{X}_{euler} = \mathbf{X} + d_1 \cdot \Delta t$.
10:     *// Corrector Step*
11:     $\mathcal{C}'_{cond} = \text{MLP}(\text{GraphBackbone}(\mathbf{X}_{euler}, \mathcal{C}_e, t_{next})) + t_{next_{emb}}$.
12:     $d_2 = -0.5 \cdot g(t_{next})^2 \cdot \text{DiT}(\mathbf{X}_{euler}, \mathcal{C}'_{cond})$.
13:     $\mathbf{X} \leftarrow \mathbf{X} + 0.5 \cdot (d_1 + d_2) \cdot \Delta t$.
14: **end for**
15: **Finalize:** Apply Tweedie projection to $t = 0$.
16: **Return** $\hat{\mathbf{X}}_0$.

---

*Table 5.* Hyperparameter Configuration for FLAG

| Category | Parameter | Value | Description |
|----------|-----------|-------|-------------|
| **Graph Backbone** | Hidden Dim ($C$) | 384 | Node/Edge feature dimension |
| | Layers / Heads | 6 / 8 | Depth and attention heads |
| | Condition Dim | 1024 | UNI visual feature size |
| | Edge Dim | 2 | Distance + Visual Similarity |
| **DiT** | Hidden Dim | 384 | Transformer width |
| | Layers / Heads | 12 / 6 | Transformer depth |
| | MLP Ratio | 4.0 | Feedforward expansion |
| | Gene Dim ($d_{gene}$) | 512 | Initial gene projection size |
| | Align Layer | 8 | Layer index used for alignment |
| **Training** | Optimizer | AdamW | with Weight Decay 0.01 |
| | Learning Rate | 1e-4 | Constant schedule |
| | Grad Clip | 1.0 | Gradient clipping norm |
| | SDE Limits | [0.01, 10.0] | $\sigma_{min}, \sigma_{max}$ |

## H. Extensive Experimental Analysis & Ablations

In this section, we provide a comprehensive analysis to validate the robustness, scalability, and efficiency of our proposed method. We cover the verification of foundation model priors (Sec. H.1), effects of alignment on different dit layer (Sec. H.2), additional experiments on gene dimensional analysis (Sec. H.3), training dynamics (Sec. H.4), uncertainty quantification (Sec. H.5), inference efficiency (Sec. H.6), training cost analysis (Sec. H.7), ablations on spatial priors (Sec. H.8), additional downstream evaluations on the DLPFC (Maynard et al., 2021) dataset (Sec. H.9), and supplementary case studies on structural fidelity (Sec. H.10).

### H.1. Verification of Foundation Model Priors

To investigate the impact of different Gene Foundation Models (GFMs) as structural priors, we conduct an ablation study in an isolated setting. Specifically, we **remove the Spatial Graph Backbone** from the FLAG framework to decouple the influence of spatial message passing, thereby focusing solely on the contribution of the GFM to feature alignment and gene-gene correlation recovery.

We compare four experimental settings on the KIDNEY dataset:

1. **No GFM**: The gene embeddings are randomly initialized and learned from scratch without any pre-trained biological knowledge.

2. **w/ scGPT**: We utilize scGPT embeddings as the prior.

3. **w/ CellPLM**: We employ CellPLM as the prior. Note that since CellPLM produces cell-level embeddings, we apply *average pooling along the gene dimension* to align its output with our target gene embedding space.

4. **w/ Geneformer (Ours)**: Our default setting using Geneformer token embeddings.

*Table 6.* **Ablation of Foundation Model Priors (w/o Graph Backbone).** Performance comparison of different GFM initialization strategies on the KIDNEY dataset on top-200 HMHVG genes panel. All variants are trained without the spatial graph backbone to independently assess the effect of the gene prior.

| Prior Strategy | PCC ↑ | MSE ↓ | GSC ↑ |
| --- | --- | --- | --- |
| No GFM | 0.3225 | 1.4258 | 0.8050 |
| w/ scGPT | 0.3364 | 1.3908 | **0.8743** |
| w/ CellPLM[*] | 0.3245 | 1.4102 | 0.8466 |
| **w/ Geneformer** | **0.3457** | **1.3527** | 0.8594 |

[*]For CellPLM, gene-dimension pooling is applied for alignment.

As shown in Table 6, the *No GFM* baseline yields the lowest PCC and GSC score, indicating that without pre-trained priors, the diffusion model struggles to capture the complex gene-gene co-expression networks solely from image features. Among the pre-trained models, while scGPT and CellPLM improve performance over the baseline, **Geneformer** achieves the best balance between PCC and GSC.

## H.2. GFM Alignment on Different Dit Layer

*Table 7.* **Layer alignment ablation.** We align FLAG's diffusion representation to frozen GFM gene embeddings extracted from different GFM layers ($\ell \in \{2, 4, 8, 12\}$), keeping all other settings identical. Results are reported across slides on KIDNEY with Top-200 HMHVG genes.

| Alignment target | PCC ↑ | GSC ↑ |
| --- | --- | --- |
| No alignment | 0.3225 | 0.8050 |
| Align dit-layer $\ell=2$ | 0.3280 | 0.8254 |
| Align dit-layer $\ell=4$ | 0.3346 | 0.8426 |
| Align dit-layer $\ell=8$ | **0.3457** | **0.8594** |
| Align to dit-layer $\ell=12$ | 0.3402 | 0.8506 |

Table 7 shows that gene-gene structure recovery (GSC) depends more strongly on the alignment depth $\ell$ than pointwise accuracy (PCC). Aligning to an intermediate layer (typically $\ell=8$) yields the best GSC, while shallower layers ($\ell=2, 4$) or deeper layers ($\ell=12$) provide smaller gains.

Intermediate GFM layers likely capture more transferable relational gene semantics than very shallow layers, and aligning the DiT features to this space can act as a weak prior on the correlation geometry of the predicted expressions, directly benefiting GSC. By contrast, shallow-layer alignment may be dominated by lower-level statistics with weaker global gene-gene organization, whereas top-layer alignment is most strongly shaped by the GFM pretraining objective and may introduce an objective-induced bias that over-constrains adaptation to spatial gene expression regression. Overall, these observations support using intermediate-layer alignment as a practical compromise between leveraging GFM priors and preserving downstream flexibility.

## H.3. Additional Experiments on Gene Dimension Curse

We extend the gene dimension analysis to the KIDNEY dataset to evaluate generalizability. We compare **FLAG** against *Joint Node-Edge Diffusion* and *Node-Only Diffusion* across gene panel sizes $G \in \{50, 100, 200, 400\}$.

*Table 8.* **Gene Dimension Experiments on KIDNEY Dataset.** Performance comparison across varying gene panel sizes ($G$). FLAG demonstrates superior stability, particularly in structural metrics (GSC and SSC) at higher dimensions ($G = 400$), whereas baselines exhibit significant degradation.

| Metric | Method | Number of Genes ($G$) | | | |
|---|---|---|---|---|---|
| | | 50 | 100 | 200 | 400 |
| PCC ↑ | Node-Only Diffusion | 0.5205 | 0.3647 | 0.1853 | 0.0131 |
| | Joint Node-Edge Diffusion | 0.5350 | 0.3461 | 0.1497 | 0.0016 |
| | FLAG | **0.5524** | **0.4873** | **0.3917** | **0.3082** |
| MSE ↓ | Node-Only Diffusion | 1.7721 | 2.5813 | 2.9257 | 16.6905 |
| | Joint Node-Edge Diffusion | 1.8564 | 2.8495 | 2.9835 | 19.8750 |
| | FLAG | **1.1921** | **1.1384** | **1.2112** | **1.9317** |
| GSC ↑ | Node-Only Diffusion | 0.6790 | 0.6295 | 0.3282 | 0.0401 |
| | Joint Node-Edge Diffusion | 0.7509 | 0.7101 | 0.4358 | 0.0603 |
| | FLAG | **0.7735** | **0.8299** | **0.8713** | **0.8603** |
| SSC ↑ | Node-Only Diffusion | 0.4230 | 0.3453 | 0.2370 | 0.1301 |
| | Joint Node-Edge Diffusion | **0.4284** | 0.2413 | 0.1064 | 0.0509 |
| | FLAG | 0.4229 | **0.4098** | **0.3409** | **0.3708** |

At lower dimensions ($G \in \{50, 100\}$), Joint Node-Edge Diffusion consistently outperforms Node-Only Diffusion in terms of GSC (e.g., 0.7509 vs. 0.6790 at $G = 50$). This supports the hypothesis that explicitly modeling the gene-gene co-evolution (edges) is beneficial for capturing biological correlations.

However, FLAG maintains remarkably stable GSC scores even at $G = 400$. This suggests that GFM effectively acts as a semantic anchor. By leveraging pretrained knowledge, the FLAG regularizes the gene-gene relationships, preventing the structural degradation typically observed when learning correlations from scratch in high-dimensional regimes.

Regarding SSC, both Node-Only Diffusion and Joint Node-Edge Diffusion exhibit severe degradation at $G = 400$, indicating a loss of spatial tissue patterns. FLAG preserves spatial fidelity, validating the effectiveness of our decoupled Spatial Graph Backbone in handling spatial consistency independently of the gene dimension size.

### H.4. Training Dynamics

To investigate the optimization characteristics of different components, we analyze the validation performance curves during the training process. Figure 9 visualizes the evolution of the Pearson Correlation Coefficient (PCC) on the HER2ST dataset over 3,000 epochs for three variants: (1) **FLAG (Ours)**, (2) FLAG **w/o GFM Alignment** (removing the foundation model prior), and (3) FLAG **w/o Graph Encoder** (removing the spatial message passing).

**Analysis of Convergence Behaviors:** As illustrated in Figure 9, the training dynamics reveal distinct distinct optimization patterns:

- **Premature Saturation of w/o Spatial Graph Encoder Model (Blue Curve):** The *w/o Graph Encoder* variant exhibits the sharpest initial rise (Epoch 0-300), suggesting that local image features are learned quickly. However, it plateaus significantly earlier than other methods. This indicates that relying solely on visual features without spatial context limits the model's capacity to resolve complex expression patterns.

- **Cold Start without Priors (Orange Curve):** The *w/o GFM Alignment* variant demonstrates a noticeably slower convergence rate compared to the w/o Graph Encoder experiment. Unlike the w/o Graph Encoder, this variant must simultaneously learn feature representations and spatial message passing dynamics. Learning these complex spatial relations imposes a heavier optimization burden in the early stages, resulting in a "slow start." However, once the spatial patterns are learned, it eventually surpasses the node-only baseline.

- **Optimal Convergence of FLAG (Green Curve):** While the full **FLAG** model also incorporates the complex graph structure, it converges faster than the *w/o GFM* variant. This indicates that the **GFM prior** acts as an effective semantic

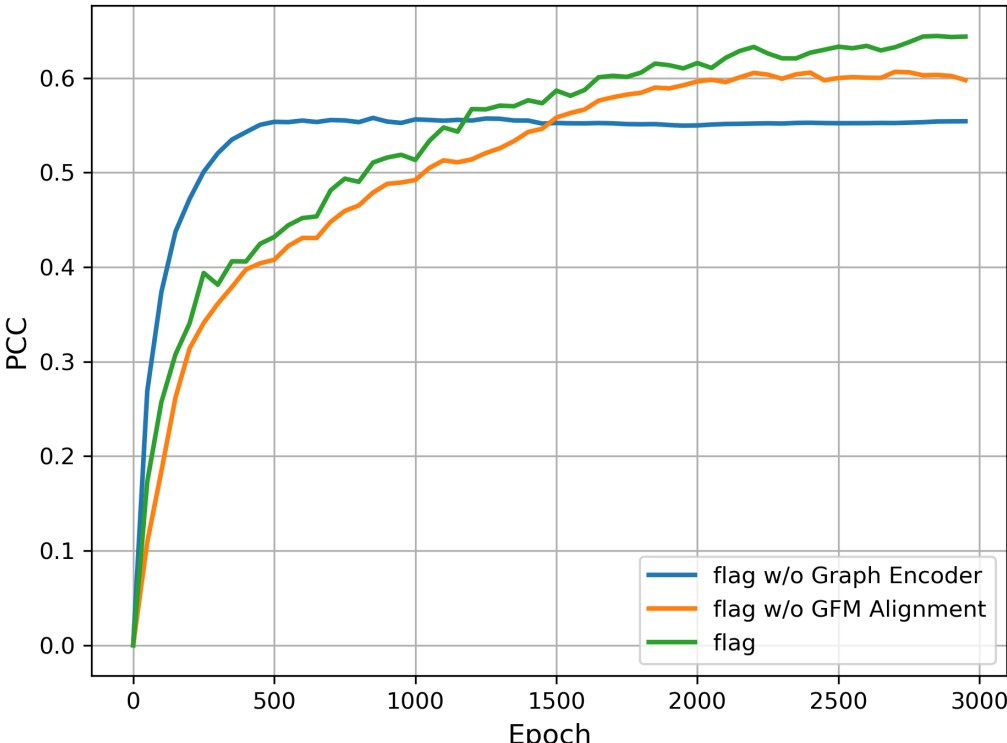

*Figure 9.* **Training Dynamics on HER2ST Dataset HMHVG-200 genes panel.** The curves depict the validation PCC over training epochs.

anchor, providing a "warm start" for the gene embeddings. The synergy between the structural capability of the graph and the semantic guidance of the FM allows FLAG to navigate the complex optimization landscape efficiently.

### H.5. Uncertainty and Sampling Variance

Generative models, such as Stem and STFlow, inherently introduce stochasticity during the inference process. For clinical applications, it is critical to ensure that this randomness does not compromise the reproducibility of the predictions.

To evaluate this, we conducted a stability analysis on the HER2ST dataset. We performed $N = 5$ independent sampling runs for FLAG and two leading generative baselines: Stem and STFlow. We report the Mean ($\mu$) and Standard Deviation ($\sigma$) across four key metrics to quantify uncertainty. In terms of gene-wise prediction (PCC and MSE), FLAG exhibits stability comparable to the baselines.

*Table 9.* **Stability Comparison with SOTA Generative Methods.** Results are averaged over 5 independent runs on HER2ST HMHVG-200 genes panel.

| Method | PCC ($\mu \pm \sigma$) | MSE ($\mu \pm \sigma$) | GSC ($\mu \pm \sigma$) | SSC ($\mu \pm \sigma$) |
|---|---|---|---|---|
| Stem | $0.5772 \pm 0.0034$ | $0.9535 \pm 0.0121$ | $0.8322 \pm 0.0145$ | $0.3810 \pm 0.1100$ |
| STFlow | $0.7058 \pm 0.0013$ | $0.6769 \pm 0.0026$ | $0.7890 \pm 0.0022$ | $0.2890 \pm 0.0176$ |
| FLAG | $0.6835 \pm 0.0067$ | $0.7342 \pm 0.0123$ | $0.8926 \pm 0.0045$ | $0.6386 \pm 0.0405$ |

### H.6. Inference Efficiency

We selected one representative test slide from each of the three datasets (HER2ST, KIDNEY, and PRAD) to cover a wide range of spatial dimensions. We recorded the inference time (seconds) and peak GPU Memory Usage (GB) during the sampling process. All measurements were conducted on a single NVIDIA H800 GPU with a batch size of 1 (processing one

WSI at a time). As shown in Table 10, the computational cost of FLAG is well within the acceptable range for practical deployment.

*Table 10.* **Inference Efficiency across Different WSI Scales.** We report the total number of spots ($N$), inference time, and peak memory usage for representative slides from each dataset.

| Dataset | # Spots ($N$) | Inference Time (s) | Peak Memory (GB) |
|---------|---------------|--------------------|------------------|
| HER2ST  | 293           | 24.39              | 2.79             |
| KIDNEY  | 317           | 26.81              | 3.10             |
| PRAD    | 1413          | 143.65             | 3.38             |

## H.7. Training Cost Analysis

We also provide a detailed training cost analysis. Our training process is highly efficient as it leverages an offline pre-computation strategy. Both the Pathology Encoder (UNI) and the Gene Foundation Model (Geneformer) are strictly frozen during training. For a typical cohort like HER2ST (comprising ~14,000 spots), extracting visual features via UNI takes approximately 30 minutes, and computing genomic representation alignment via Geneformer (HMHVG-200 set) takes about 25 minutes. These are one-time costs, and the resulting embeddings are cached for all subsequent training.

*Table 11.* **Training Efficiency and Resource Consumption on the HER2ST Dataset (Single NVIDIA H800 GPU).**

| Method | Training Time (s/epoch) | Peak GPU Memory (GB) |
|--------|-------------------------|----------------------|
| HisToGene | 25 | 4.0 |
| Stem | 30 | 4.0 |
| BLEEP | 320 | 2.0 |
| STFlow | 20 | 1.5 |
| TRIPLEX | 40 | 32.0 |
| **FLAG (Ours)** | 35 | 4.5 |

During active training, only the diffusion backbone is optimized. As shown in Table 11, FLAG's resource footprint on a single NVIDIA H800 GPU is completely comparable to existing baselines, requiring only 4.5 GB of peak memory and 35 seconds per epoch. Trained for 1,000 epochs until convergence, FLAG requires approximately 9.7 hours of total active training time, which perfectly aligns with the computational budget of other generative models (e.g., Stem).

## H.8. Ablations on Spatial Priors

To verify the robustness, interpretability, and effectiveness of our spatial graph construction, we conducted extensive sensitivity analyses and ablations on the HER2ST dataset.

**SSC Sensitivity and the Rationale for $k = 8$.** The parameter $k = 8$ is purely an evaluation hyperparameter for SSC (Moran's I neighborhood). It does not alter predicted expressions when calculating pointwise metrics (PCC, MSE). Recalculating SSC on fixed HER2ST predictions with $k \in \{6, 8, 12\}$ shows high stability (Table 12). This confirms that our structural fidelity robustly reflects true generative quality rather than metric artifacts.

*Table 12.* **Sensitivity validation of the SSC evaluation parameter $k$ on the HER2ST dataset.**

| Metric | $k = 6$ | $k = 8$ (Default) | $k = 12$ |
|--------|---------|-------------------|----------|
| SSC ($\uparrow$) | 0.6365 | 0.6386 | 0.6290 |

**Topology Prior: Interpretability and Robustness of $\sigma = 224$.** The choice of $\sigma = 224$ is a physical heuristic. Graph coordinates are in image pixels, and $224 \times 224$ is the exact bounding box of a single spot's patch. Setting $\sigma = 224$ in $W_{ij} = \exp(-d_{ij}^2/\sigma^2)$ aligns distance decay naturally with a spot's visual field. Varying $\sigma$ (Table 13) shows that $\sigma = 224$ yields optimal SSC, while overall performance remains highly stable.

*Table 13.* **Robustness validation for the distance-kernel length-scale $\sigma$ on the HER2ST dataset.**

| Metric | $\sigma = 112\,(0.5\times)$ | $\sigma = 224\,(\text{Default, }1\times)$ | $\sigma = 448\,(2\times)$ |
|---|---|---|---|
| **PCC** ($\uparrow$) | 0.6816 | 0.6835 | 0.6794 |
| **MSE** ($\downarrow$) | 0.7385 | 0.7342 | 0.7406 |
| **GSC** ($\uparrow$) | 0.8848 | 0.8926 | 0.8914 |
| **SSC** ($\uparrow$) | 0.6412 | 0.6386 | 0.6364 |

**Topology Prior: Distance vs. Histology Channels and Normalization.** Both edge channels in FLAG are inherently bounded, avoiding the need for manual scaling. Specifically, the distance channel uses an RBF kernel $W_{\text{dist}} \in (0, 1]$, and the histology channel uses cosine similarity $W_{\text{hist}} \in [-1, 1]$. Operating within this $[-1, 1]$ range, they are simply concatenated. The Graph Encoder automatically learns the optimal fusion without relative scaling. Table 14 confirms that integrating both channels yields the best spatial prior.

*Table 14.* **Ablation of edge channels within the Spatial Graph Encoder on HER2ST.**

| Edge Prior Provided | PCC ($\uparrow$) | GSC ($\uparrow$) | SSC ($\uparrow$) |
|---|---|---|---|
| Distance-based Kernel Only | 0.6638 | 0.8726 | 0.6194 |
| Histology Similarity Only | 0.6592 | 0.8432 | 0.5874 |
| **Both (FLAG Default)** | **0.6835** | **0.8926** | **0.6386** |

### H.9. Additional Downstream Evaluations on the DLPFC Dataset

To further establish that our structure-aware metrics translate into downstream biological utility, we extended our evaluations to the DLPFC dataset (Slide 151673). Unlike HER2ST, the DLPFC dataset provides human expert layer annotations, which reflect true biological anatomy.

**Spatial Domain Identification.** To test whether the generated expression preserves spatial domain structure, we applied standard Scanpy workflows: PCA (20 PCs), neighborhood graph ($k = 15$), and Leiden clustering (resolution 0.3 to match anatomical regions). The results are evaluated directly against the human expert layer annotations. As shown in Table 15, the clustering performance based on Ground Truth (GT) expression establishes a practical upper bound (ARI 0.4814, NMI 0.6256). FLAG's predictions approach this GT upper bound significantly more closely than all baselines. These objective trends confirm that FLAG provides more reliable utility for spatial downstream tasks. Note that TRIPLEX encountered Out-Of-Memory (OOM) errors on this slide.

**DEG Consistency.** Identifying consistent Differentially Expressed Genes (DEGs) is crucial to verify whether the generated expressions capture biologically meaningful differential signals. To ensure fair comparisons, we evaluated all models within the exact same reference spatial domains (the expert-annotated layers) using a one-vs-rest Wilcoxon rank-sum test. As reported in Table 15, FLAG consistently exhibits the highest overlap rate of marker genes among the Top-20 and Top-50 DEGs against the GT reference. These results demonstrate that FLAG successfully preserves the critical localized differential signals required for authentic downstream marker discovery.

### H.10. Supplementary Case Studies

In this section, we provide additional qualitative results to substantiate the structural fidelity of FLAG.

#### H.10.1. GSC CASE STUDY

To qualitatively assess the recovery of functional gene modules, we visualize the gene-gene correlation matrices for two critical renal pathways: Hypoxia (reflecting metabolic stress response) and Epithelial Mesenchymal Transition (EMT) (a key driver of renal fibrosis) on KIDNEY Dataset. Figure 10 compares the correlation structures generated by FLAG against the Ground Truth and representative baselines.

*Table 15.* **Quantitative Downstream Evaluations on DLPFC (Slide 151673).** Spatial Domain Identification (ARI ↑, NMI ↑) is evaluated directly against human expert annotations. DEG consistency (Top-20 ↑, Top-50 ↑) is computed within these expert-annotated domains. GT Expression serves as the upper bound for clustering.

| Method | Spatial Domain Clustering | | DEG Overlap Ratio | |
|---|---|---|---|---|
| | ARI ↑ | NMI ↑ | Top-20 ↑ | Top-50 ↑ |
| *GT (Upper Bound)* | *0.4814* | *0.6256* | - | - |
| HisToGene | 0.0785 | 0.1854 | 37.86% | 59.43% |
| Stem | 0.1136 | 0.1954 | 64.86% | 74.86% |
| BLEEP | 0.1208 | 0.2748 | 54.29% | 66.67% |
| STFlow | 0.2453 | 0.3986 | 55.71% | 68.48% |
| TRIPLEX | OOM | OOM | OOM | OOM |
| **FLAG (Ours)** | **0.3654** | **0.5357** | **66.57%** | **79.71%** |

### H.10.2. SPATIAL PATTERN RECOVERY OF MARKER GENES

To further validate the model's capability in reconstructing diverse spatial textures, we visualize the spatial expression maps of three representative marker genes in the KIDNEY dataset (PODXL, LRP2, VIM). Figure 11 presents the comparison between Ground Truth, FLAG, and baselines.

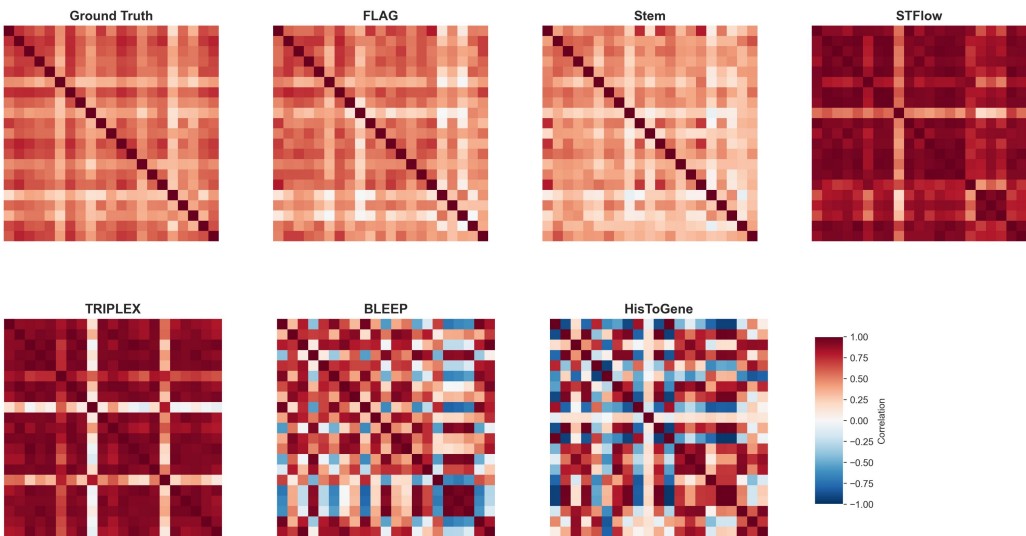

(a) Epithelial Mesenchymal Transition Pathway Heatmap

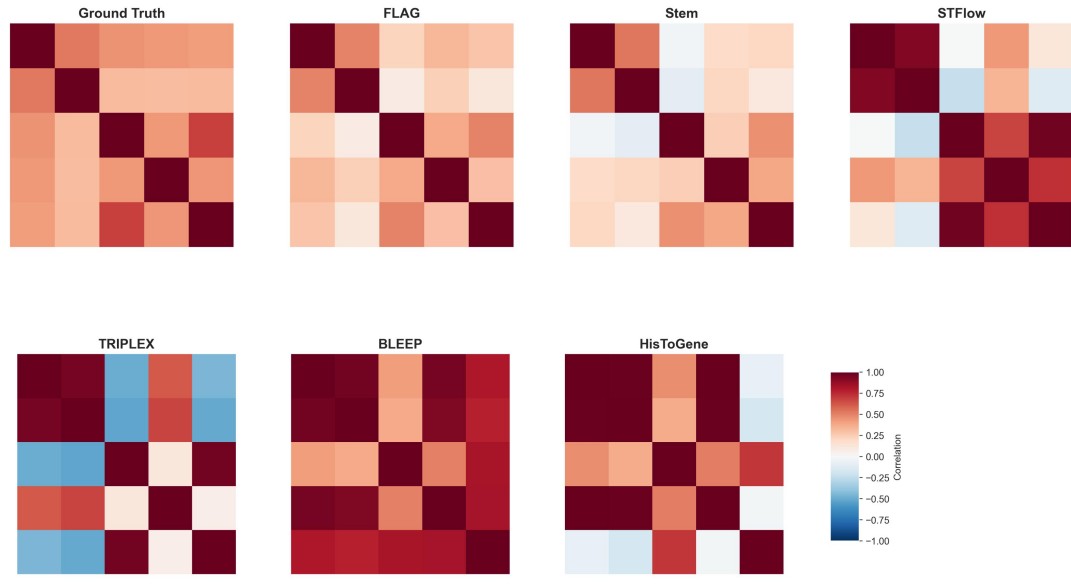

(b) Hypoxia Pathway Heatmap

*Figure 10.* **Qualitative Comparison of Gene Regulatory Networks on KIDNEY Dataset. (a) EMT Pathway**. **(b) Hypoxia Pathway**.

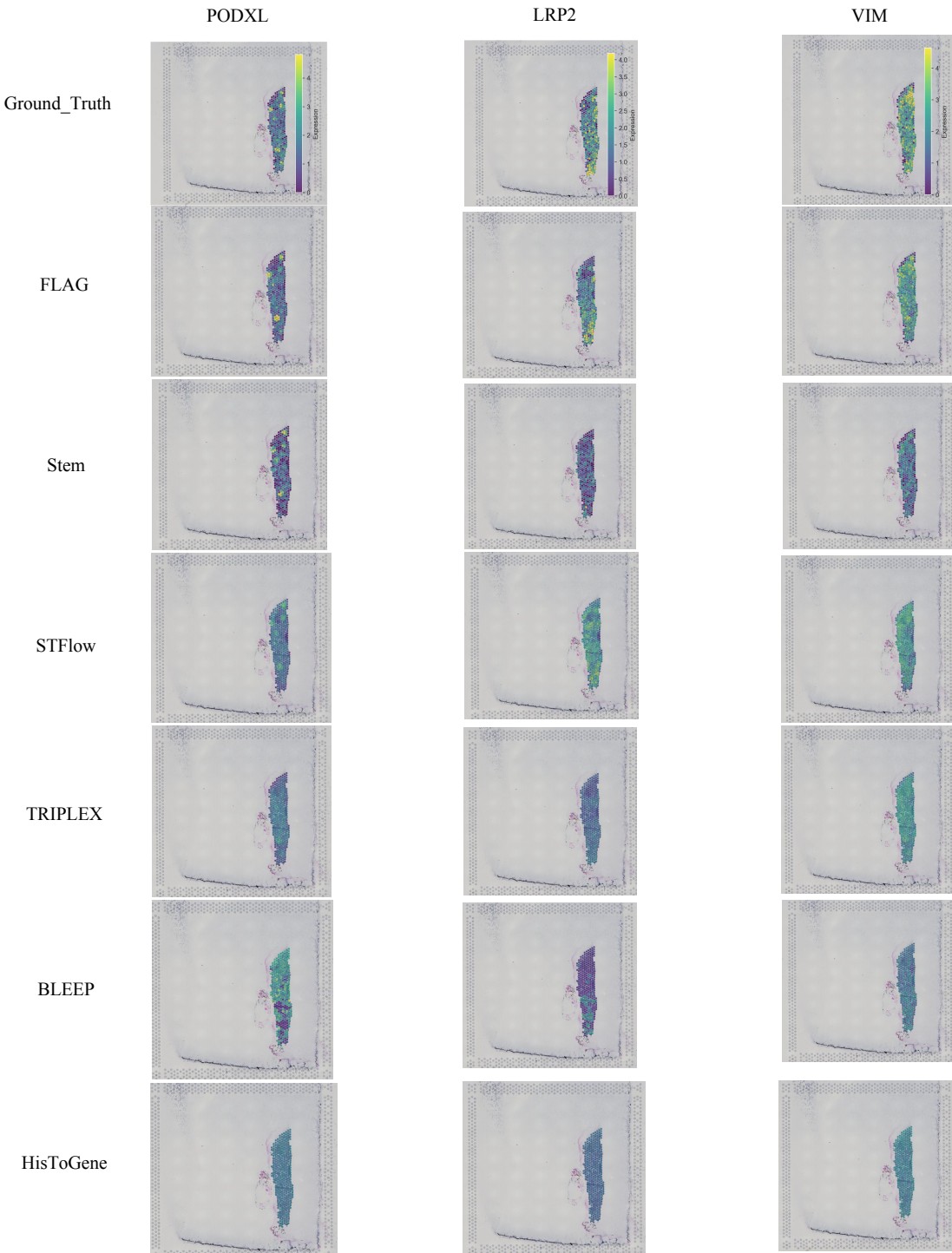

*Figure 11.* **Spatial Expression Recovery on KIDNEY Dataset.** Visual comparison of three marker genes representing distinct spatial textures: **PODXL**, **LRP2**, and **VIM**.

