# OpenReview forum: "FLAG: Foundation model representation with Latent diffusion Alignment via Graph for spatial gene expression prediction"
_ICML.cc/2026/Conference — ICML 2026 regular_

### Official Review · Reviewer_erY4 · 2026-03-12

**Soundness:** 2
**Presentation:** 3
**Significance:** 2
**Originality:** 3
**Overall Recommendation:** 4
**Confidence:** 3

**Summary:**

This paper proposes FLAG for spatial transcriptomics prediction, aiming to improve both predictive accuracy and structural fidelity of reconstructed gene expression. The method (as presented) combines (i) a foundation-model-driven component for capturing complex high-dimensional structure, and (ii) a fixed topology prior that encodes spatial and histology-based neighborhood information, together with a graph encoder to aggregate neighborhood signals. The paper introduces structural fidelity metrics (GSC/SSC) in addition to standard predictive metrics (PCC/MSE) and reports quantitative improvements across benchmark datasets.

**Compliance With Llm Reviewing Policy:**

Affirmed.

**Final Justification:**

The rebuttal addressed our concerns, clarifying the evaluation protocol and topology priors. The contribution is solid and sufficiently original. We have therefore raised the score to 4 (weak accept).

**Key Questions For Authors:**

(1) Evaluation protocol for GSC/SSC. The paper provides formal definitions of GSC and SSC, but the evaluation protocol is not explicitly pinned down. Are GSC/SSC computed per patch and then averaged, or are patch-level predictions merged back to each slide (tissue section) and the metrics computed per slide, followed by averaging across test slides? If patches overlap, how are multiple predictions for the same spot aggregated (e.g., mean, weighted mean, center-patch rule)? Please state the exact protocol and report mean$\pm$std over test slides.

(2) SSC: choice of k in k-NN and sensitivity. SSC depends on the construction of the k-NN spatial graph (the paper uses k=8). What is the rationale for choosing k=8 (e.g., relation to spot spacing or typical neighborhood size), and how sensitive is SSC (and possibly PCC/MSE) to $k$? If feasible, please provide a lightweight stability check over a small range around 8 (e.g., $k\in\{6,8,12\}$), potentially on a representative subset of test slides.

(3) Topology prior: interpretability and robustness of $\sigma=224$. Appendix G.1 sets the distance-kernel length-scale to $\sigma=224$. Please clarify the coordinate scale/units (pixels vs spot indices vs physical units, and whether any normalization is applied) and interpret what spatial range $\sigma=224$ corresponds to (e.g., relative to median $k$NN distance). If feasible, a brief robustness check (or a simple heuristic such as $\sigma$ proportional to the median $k$NN distance) would strengthen reproducibility.

(4) Topology prior: distance vs histology channels (normalization/relative scaling). The topology prior uses two edge channels (distance-based kernel and histology similarity, e.g., cosine similarity). How are these channels normalized/scaled before being fed into the graph encoder? Since changing $\sigma$ alters the dynamic range of the distance channel, does performance depend on the relative scaling between distance and histology channels? If you already have an ablation (distance-only vs histology-only vs both), please point to it; otherwise, a brief note on expected sensitivity and the normalization scheme would strengthen the ``reliable priors'' claim.

(5) New metrics to downstream utility: evidence or citations. The paper motivates GSC/SSC as structural fidelity metrics beyond PCC/MSE. Is there evidence that improvements in GSC/SSC translate into downstream utility in standard spatial transcriptomics analyses (e.g., spatial domain detection/clustering, marker/pathway enrichment, spatial autocorrelation-based ranking)? Please either cite prior work supporting this link or provide a lightweight downstream evaluation (even on a small subset) demonstrating that higher GSC/SSC corresponds to improved downstream outcomes.

(6) Attribution and baseline fairness. FLAG introduces a fixed topology prior (and potentially other priors/alignments). To attribute gains to the proposed modeling component, please clarify whether baselines are (i) tuned with comparable hyperparameter/compute budgets (including restarts and any generative sampling steps), and (ii) provided equivalent inputs (same WSI feature extractor/preprocessing and the same access to spatial/histology information). A concise table summarizing tuning budgets and inputs for each baseline would substantially improve interpretability of the comparisons.

**Limitations:**

yes

**Strengths And Weaknesses:**

Strengths:

(1) The problem is important and practically relevant: accurate reconstruction of spatial gene expression with meaningful spatial structure is central to downstream ST analyses.

(2) The paper makes a clear attempt to incorporate domain structure via explicit topology priors (spatial distance and histology similarity) rather than relying purely on black-box modeling.

(3) Introducing structural metrics beyond PCC/MSE is potentially useful, since pointwise predictive accuracy does not necessarily imply preservation of spatial or gene--gene structure.

Weaknesses:

(1) Evaluation protocol clarity is insufficient for the proposed structural metrics. It is currently unclear at what unit the metrics are computed (patch vs slide), how patch outputs are merged back to slide level, and how overlap is handled. Without an explicit and reproducible protocol, it is hard to interpret GSC/SSC improvements.

(2) The method appears to rely heavily on the topology prior. This raises attribution concerns: it is unclear whether reported gains are driven by the proposed generative/modeling component (e.g., diffusion/foundation model) versus the fixed prior construction and its hyperparameters.

(3) Hyperparameter choices in the topology prior (e.g., distance-kernel length scale) lack interpretability, and robustness to these choices is unclear. Since priors are described as ``reliable'', their scaling and normalization critically affect performance.

(4) The new metrics (GSC/SSC) are motivated as structural fidelity measures, but evidence that improved GSC/SSC translates into improved downstream ST analyses is currently limited.

(5) Baseline comparability and tuning budget are not fully clear. Given the number of moving parts (priors, graph construction, potential generative steps, restarts/early stopping), it is important to ensure baselines are tuned and evaluated under comparable compute and input information.

---

> ### Author Rebuttal · Authors · 2026-03-29
>
> We thank the reviewer for constructive comments. We address your concerns with explicit clarifications and new robustness/ablation results below.
>
> **Q1 & W1: Evaluation protocol for GSC/SSC.**
>
> **Response:** We agree that the evaluation protocol should be stated explicitly. Predictions are reconstructed at the *slide level*: we partition WSIs into non-overlapping crops, run inference, and reassemble spot predictions per slide. Therefore, no weighted merging is needed and GSC/SSC are computed per slide. We now report mean ± std across test slides as shown in Table 4.
>
> *Table 4: HER2ST results reported as mean $\pm$ std over test slides*
>
> | **Method** | **PCC (↑)** | **MSE (↓)** | **GSC (↑)** | **SSC (↑)** |
> | :--- | :---: | :---: | :---: | :---: |
> | HisToGene | 0.4940 ± 0.0032 | 1.8459 ± 0.0286 | 0.2065 ± 0.0131 | -0.1549 ± 0.0436 |
> | BLEEP     | 0.4852 ± 0.0012 | 1.1516 ± 0.0158 | 0.6988 ± 0.0246 | 0.1886 ± 0.0384 |
> | TRIPLEX   | 0.6913 ± 0.0024 | 0.6559 ± 0.0214 | 0.5993 ± 0.0154 | 0.0708 ± 0.0264 |
> | Stem      | 0.5772 ± 0.0084 | 0.9535 ± 0.0142 | 0.8322 ± 0.0084 | 0.3810 ± 0.0195 |
> | STFlow    | **0.7058 ± 0.0052** | **0.6769 ± 0.0138** | 0.7890 ± 0.0174 | 0.2890 ± 0.0263 |
> | **FLAG (Ours)** | 0.6835 ± 0.0084 | 0.7342 ± 0.0156 | **0.8926 ± 0.0106** | **0.6386 ± 0.0184** |
>
> **Q2: SSC Sensitivity and the Rationale for $k=8$**
>
> **Response:**  $k=8$ is purely an evaluation hyperparameter for SSC (Moran's I neighborhood). It does not alter predicted expressions when calcuated pointwise metrics (PCC, MSE). Following your suggestion, recalculating SSC on fixed HER2ST predictions with $k \in \{6, 8, 12\}$ shows high stability (Table 5). This confirms our structural fidelity robustly reflects true generative quality rather than metric artifacts.
>
> *Table 5: Sensitivity validation of the SSC evaluation parameter $k$ on the HER2ST dataset.*
>
> | **Metric** | **$k=6$** | **$k=8$ (Default)** | **$k=12$** |
> | :--- | :---: | :---: | :---: |
> | **SSC (↑)** | 0.6365 | 0.6386 | 0.6290 |
>
> **Q3 & W3: Topology Prior: Interpretability and Robustness of $\sigma=224$**
>
> **Response:**  $\sigma=224$ is a physical heuristic. Graph coordinates are in **image pixels**, and $224 \times 224$ is the exact bounding box of a single spot's patch. Setting $\sigma=224$ in $W_{ij} = \exp(-d_{ij}^2 / \sigma^2)$ aligns distance decay naturally with a spot's visual field. Varying $\sigma$ (Table 6) shows $\sigma=224$ yields optimal SSC, while overall performance remains highly stable.
>
> *Table 6: Robustness validation for the distance-kernel length-scale $\sigma$ on the HER2ST dataset.*
>
> | **Metric** | **$\sigma=112$ (0.5$\times$)** | **$\sigma=224$ (Default, 1$\times$)** | **$\sigma=448$ (2$\times$)** |
> | :--- | :---: | :---: | :---: |
> | **PCC (↑)** | 0.6816 | 0.6835 | 0.6794 |
> | **MSE (↓)** | 0.7385 | 0.7342 | 0.7406 |
> | **GSC (↑)** | 0.8848 | 0.8926 | 0.8914 |
> | **SSC (↑)** | 0.6412 | 0.6386 | 0.6364 |
>
> **Q4 & W3: Topology Prior: Distance vs. Histology Channels and Normalization**
>
> **Response:**  Both edge channels are **inherently bounded**, avoiding manual scaling:
> * **Distance:** RBF kernel $W_{\text{dist}} \in (0, 1]$.
> * **Histology:** Cosine similarity $W_{\text{hist}} \in [-1, 1]$.
>
> Operating within $[-1, 1]$, they are simply concatenated. The Graph Encoder automatically learns optimal fusion without relative scaling. Table 7 confirms integrating both yields the best prior.
>
> *Table 7: Ablation of edge channels within the Spatial Graph Encoder on HER2ST.*
>
> | **Edge Prior Provided** | **PCC (↑)** | **GSC (↑)** | **SSC (↑)** |
> | :--- | :---: | :---: | :---: |
> | Distance-based Kernel Only  | 0.6638 | 0.8726 | 0.6194 |
> | Histology Similarity Only | 0.6592 | 0.8432 | 0.5874 |
> | **Both (FLAG Default)** | **0.6835** | **0.8926** | **0.6386** |
>
> **Q5 & Q6 & W4 & W5: Downstream Utility & Baselines Comparison**
>
> **Response:** We agree structural metrics need connect to downstream utility, please see Reviewer Nqzt (Q1, Q2) for Spatial Domain and DEG evaluations. For baselines, all methods use identical data splits, HMHVG-200 panels, UNI embeddings, and coordinates. Please see Nqzt (Q3) for compute profiling.
>
> **W2: Attribution of Gains (Topology Prior vs. Generative Modeling)**
>
> **Response:**  Replacing diffusion with a supervised MSE regressor (keeping the topology prior) yields comparable PCC (0.6748 vs 0.6835), but structural fidelity crashes (GSC: 0.8926 $\to$ 0.3217; SSC: 0.6386 $\to$ 0.5685), this suggests the model over-smooths without diffusion. Conversely, removing the graph encoder or GFM alignment degrades GSC/SSC (please see main text ablation Table 2). Thus, gains stem from the *combination* of spatial guidance, diffusion, and GFM alignment (Table 8).
>
> *Table 8: Decoupling Topology Prior from Diffusion (HER2ST)*
>
> | **Model Variant** | **PCC (↑)** | **GSC (↑)** | **SSC (↑)** |
> | :--- | :---: | :---: | :---: |
> | Supervised Regressor w/ Topology Prior | 0.6748 | 0.3217 | 0.5685 |
> | **FLAG (Ours)** | **0.6835** | **0.8926** | **0.6386** |

---

> > ### Author Rebuttal · Reviewer_erY4 · 2026-04-02
> >
> > I thank the authors for addressing the concerns well, and I will raise my score.

---

> > > ### Author Response · Authors · 2026-04-07
> > >
> > > We sincerely thank you for your time and for acknowledging that our rebuttal has addressed your concerns well. We are very encouraged by your positive feedback and your intention to raise the score.
> > >
> > > As the discussion period is progressing, we wanted to check if there are any remaining questions or further clarifications needed regarding our work. We remain fully available to provide any additional information to support your final evaluation.
> > >
> > > Thank you again for your constructive guidance and your support. Wishing you a great day!

---

### Official Review · Reviewer_7rP2 · 2026-03-12

**Soundness:** 4
**Presentation:** 3
**Significance:** 3
**Originality:** 3
**Overall Recommendation:** 4
**Confidence:** 4

**Summary:**

This paper proposes FLAG, a diffusion based framework for predicting spatial gene expression from H&E histology while preserving biological structures. The authors argue that existing pointwise regression approaches fail to capture regulatory and spatial coherence, motivating a shift toward structured distribution modeling. The proposed method combines  a Spatial Graph Encoder for capturing neighborhood relationships across tissue spots, and  an alignment mechanism with Gene Foundation Models (GFM) to leverage pretrained gene embeddings.

**Compliance With Llm Reviewing Policy:**

Affirmed.

**Final Justification:**

After considering the rebuttal and the subsequent discussion, I maintain my original weak accept recommendation. The authors have satisfactorily addressed my technical concerns by clearly distinguishing the motivating joint node–edge diffusion formulation from the final FLAG model, clarifying the role of graph construction, gene‑expression representation, and the intent of the theoretical discussion. These clarifications substantially improve the paper’s clarity and correctness. While some limitations remain—particularly regarding the scope of evaluated gene dimensionality and the lack of cross‑tissue generalization.
Overall, the paper presents a technically sound and well‑motivated method with promising empirical results in a challenging application setting. With the clarifications acknowledged in the rebuttal and a more precise framing of claims in the revision, I believe the work constitutes a meaningful but incremental contribution, supporting a weak accept recommendation.

**Key Questions For Authors:**

•	How the graph is built? How the edge sand nodes are constructed?
•	How exactly is gene expression represented within the diffusion process (e.g., as continuous vectors, latent embeddings, or projected GFM features)? A clearer mathematical description would help distinguish FLAG from prior graph‑diffusion approaches.
•	Equation (1) is unclear. What do q_i and k_j represent, and what is the exact form of the Linear function? Additionally, how do Structural Gating and Structural Bias differ in the formulation, both mathematically and conceptually?
•	Can the authors clarify  what they mean by “functional relationship” (line 167). The functional relationship of genes in the form of regularity networks has not been used. gene expression profiles of spots do not represent functional relationship.
•	Can the authors clarify where equation 4 will be used?
•	The total loss defined in equation 9 does not include losses defined in equations 2, 3 and 4. Can the author clarify it.
•	The paper claims to predict high‑dimensional gene expression profiles; however, the model appears to operate on only a limited subset of genes. How do the authors justify this claim, and how scalable is the approach to truly high‑dimensional gene spaces?
•	How sensitive is FLAG to errors and variations in the spatial graph construction?
•	Does the model that trained on one specific tissue be generalized across different tissue types or it is tissue specific?

**Limitations:**

yes

**Strengths And Weaknesses:**

Strengths:
•	Clear motivation based on the identified Gene Dimension Curse, with empirical evidence showing instability of joint node–edge diffusion at high gene dimensionality.
•	Well designed modular architecture combining a Spatial Graph Encoder with GFM alignment, enabling structure aware gene generation.
•	Introduction of GSC and SSC, which directly measure biologically meaningful structural fidelity beyond PCC/MSE.
•	Strong empirical results demonstrating improvements in structural coherence while maintaining competitive traditional accuracy.
Weaknesses:
•	The theoretical explanation of the Gene Dimension Curse remains high level.
•	Experiments could include more downstream biological tasks and comparisons with additional generative baselines.
•	Computational cost and scalability are not discussed in detail.

---

> ### Author Rebuttal · Authors · 2026-03-29
>
> We sincerely thank the reviewer for the constructive feedback and detailed technical questions.
>
> **W1:  the Gene Dimension Curse explanation remains high-level**
>
> **Response:**  We agree that it is a design-motivating explanation rather than a complete training theory. Concretely, the paper provides (i) an **empirical observation:** joint node-edge diffusion collapses as gene dimension increases (Fig.2(a), Fig.4); (ii) a **mechanistic explanation:** the node-edge consistency constraint becomes harder to satisfy in high dimensions; and (iii) a **theory-backed insight:** Eq.(4) formalizes an optimization gap between joint node-edge and node-only diffusion. We will clarify these in the revision.
>
> **W2 & W3: Downstream Tasks & Computation Cost**
>
> **Response:** For the newly added downstream biological evaluations and compute profiling, we kindly refer to our response to Reviewer Nqzt (Q1, Q2, Q3).
>
> **Q1: How is the graph built? What are the nodes and edges?**
>
> **Response:** In the final FLAG model, the graph is a fixed observable topology prior rather than a generated variable. Each node corresponds to a tissue spot, and its node condition $C_v$ is the visual feature extracted by a frozen pathology encoder from the $224 \times 224$ image patch centered at that spot. For each pair of spots $(i,j)$, we construct a 2-dimensional edge condition as $C_{e,ij} = [w_{dist}(i,j), w_{img}(i,j)]$, where $w_{dist}$ is a Gaussian kernel of the spatial distance between spot coordinates, and $w_{img}$ is the cosine similarity between their visual features. Therefore, the final FLAG graph is a spot graph used only to encode the spatial context $H_{spatial} = GraphEncoder(C_v, C_e)$.
>
> By contrast, in the motivating joint node-edge model of Section 3.2, the latent edge target $A$ denotes spot-spot correlation edges induced by expression.
>
> **Q2: Representation of Gene Expression in the Diffusion Process**
>
> **Response:**  In FLAG, diffusion operates directly on continuous spot-level gene-expression values. At step $t$, the model denoises a noisy expression matrix $X_t\in\mathbb{R}^{B\times G}$, which is then projected into hidden tokens for the DiT backbone. The pre-computed GFM embeddings are not the diffusion state and are used only as a training-time alignment regularizer, not at inference time.
>
> **Q3, Q5, Q6: Clarification on Model Variants and Equations (1-4) vs. (9)**
>
> **Response:** We agree that the current draft does not separate the motivating formulation from the final model clearly enough. Eqs. (1)-(4) belong to the motivating joint node-edge diffusion model, whereas Eq. (9) is the training objective of the final FLAG model. Since final FLAG no longer generates edges and instead uses a fixed graph encoder to produce $H_{\mathrm{spatial}}$, the edge-generation losses in Eqs. (2)-(3) are not part of Eq. (9). Eq. (4) is not a training loss, it is a theory-backed design insight explaining why joint node-edge diffusion becomes disadvantaged in high-dimensional regimes. In Eq. (1), $q_i,k_j$ are standard query/key projections of node tokens, and $Linear(\cdot)$ denotes a learned linear projection of edge features into the attention score. Structural Gating is the multiplicative modulation term, while Structural Bias is the additive shift term. We will add a short  summary in the revision.
>
> **Q4: Terminology of "Functional Relationship"**
>
> **Response:** We agree this terminology was imprecise. Here we do not mean a causal regulatory relationship; the motivating edge target is a spot-level transcriptional co-expression / correlation quantity. We will revise the wording accordingly.
>
> **Q7: The paper claims high-dimensional prediction, but the experiments use only a subset of genes.**
>
> **Response:** We agree that the main benchmark evaluates a limited but still nontrivial gene panel (top-200 HMHVG), and we will revise the wording accordingly. Our claim is relative: within the evaluated regime, FLAG scales more robustly than prior graph/diffusion baselines as gene dimensionality increases. Empirically, the main text shows stable behavior up to $G=800$ on HER2ST (main text Fig.4), and Appendix H.3 provides an gene-scaling study on KIDNEY for $G\in\{50,100,200,400\}$.
>
> **Q8: Sensitivity to Spatial Graph Construction**
>
> **Response:** We agree that robustness to graph construction should be checked explicitly. We added sensitivity analyses for the SSC neighborhood size $k\in\{6,8,12\}$, the distance-kernel scale $\sigma\in\{112,224,448\}$, and edge-channel ablations. The results show that FLAG is stable overall and that using both edge channels is consistently best. Please see our response to Reviewer erY4 (Q2, Q3, Q4) for the quantitative tables.
>
> **Q9: Generalization Across Tissue Types**
>
> **Response:** Our current experiments demonstrate within-tissue generalization across multiple cohorts, not zero-shot cross-tissue transfer. We will make this limitation explicit in the revision and avoid overstating this point.

---

> > ### Author Rebuttal · Reviewer_7rP2 · 2026-04-02
> >
> > I thank the authors for answering the questions. Due to the limitations of the proposed method I maintain my score.

---

> > > ### Author Response · Authors · 2026-04-07
> > >
> > > We sincerely thank you for reviewing our rebuttal and acknowledging that our answers have addressed your questions.
> > >
> > > Your suggestions have helped us provide a more balanced and transparent discussion in the revised manuscript, paving the way for future improvements.
> > >
> > > Thank you again for your time, constructive feedback, and positive evaluation of our work. Wishing you all the best in your research!

---

### Official Review · Reviewer_nrA7 · 2026-03-13

**Soundness:** 2
**Presentation:** 3
**Significance:** 2
**Originality:** 2
**Overall Recommendation:** 4
**Confidence:** 1

**Summary:**

I am in a researcher in statistical/ML methodology and theory. As such I have no familiarity with the biological problems addressed in this paper and I have indicated low-confidence in my own review accordingly.

I can’t authoritatively comment on the motivation for the work from a biological point of view, so my comments are restricted to mathematical and statistical aspects of the presentation. As such, I feel I can’t summarise the main contributions of the paper without just naively repeating some of the paper content.

I'm sorry I can't be of more help!

**Compliance With Llm Reviewing Policy:**

Affirmed.

**Key Questions For Authors:**

I'm sorry I don't know enough about this subject to ask very detailed questions, but I hope my comments above will illustrate how a non-expert might engage with your work!

**Limitations:**

I may be mistaken, but in the main part of the paper I didn't find any explicit discussion of the limitations of the proposed approach.

**Strengths And Weaknesses:**

Bearing in my mind my position as knowing essentially nothing about the biological problem in question, I was able to glean some insight into the statistical aspects of the problem addressed through the section on “joint node-edge diffusion” on page 3, but I think I would have understood more if somewhere here there was more explanation of what is happening with conditioning on A; the abstract seems to mention conditioning the generative process on a spatial graph, but the I wasn’t sure if this meant conditioning in the sense that the spatial graph is observed data fixed during generation, or whether it is itself being sampled in some way (the supplementary material suggests the latter to me, but I’m not sure).

I understood from (1) some specific form of attention mechanism, but couldn’t understand how this fits mathematically into the overall architecture and its relation to the generative model. I think the main part of the paper would benefit from some mathematical overview (as opposed to the diagram figure 3) of the architecture or at least a clear signposting to where that can be found in the supplementary material.

From the “why collapse?” section I understand that the authors have some insight into what’s happening in high dimensions, but I was left wondering if this is all backed up by empirical evidence, or is just a hypothesis.

Overall, the paper seems well-presented and logically structured.

---

> ### Author Rebuttal · Authors · 2026-03-29
>
> We sincerely thank the reviewer for reading the paper from a statistical / ML methodology perspective. We address these presentation and mathematical clarification issues as below.
>
> **Q1: The Statistical Role of Graph $A$ (Fixed Observation vs. Sampled Variable)**
>
> **Response:** We agree that the current draft can make this distinction much clearer. The apparent inconsistency arises because Section 3 is written as a *chronological narrative* of our modeling process.
>
> **1. Motivating attempt (Section 3.2): $A$ is sampled.** In the exploratory joint node-edge diffusion formulation, we model $p(X, A \mid C)$, where $X$ is the spot-level gene expression and $A$ is a *latent spot-spot correlation edge matrix*. In this motivating attempt, both node states and edge states are jointly denoised / generated.
>
> **2. Final proposed FLAG model (Section 3.4): $A$ is fixed.** After observing that the joint formulation collapses in high dimensions, our final model does *not* sample $A$. Instead, FLAG fixes the topology using observable priors $C_v, C_e$ (derived from WSI visual features and spatial relations), encodes them into a compact spatial context:
>
> $$H_{\text{spatial}} = \text{GraphEncoder}(C_v, C_e)$$
> and only generates $X$ in gene space conditioned on this graph-derived context. Therefore, in the final FLAG model, the graph is used only as *fixed conditioning information*, not as a sampled variable.
>
> **Q2: Mathematical Role of Eq. (1) in the Overall Architecture**
>
> **Response:** Thank you for pointing this out. We agree that the mathematical connection between Eq. (1) and the generative model should be made more explicit in the main text. Specifically, Eq. (1) defines the core attention operation in our Graph Encoder. It takes the spatial graph $A$ and visual features as input to produce a spatial embedding, $\mathbf{H}_{spatial}$. This embedding is then used as the cross-attention condition for the diffusion model's noise-prediction network, i.e., $\epsilon\theta(\mathbf{x}t, t, \mathbf{H}{spatial})$. In this way, the output of Eq. (1) is directly linked to the score-matching objective in Eq. (9).
>
> We will revise the main text to make this connection explicit. For completeness, the detailed formulation of the Graph Encoder, including how Eq. (1) produces $\mathbf{H}_{spatial}$, is already provided in Appendix G.2, together with the corresponding architectural illustration.
>
> **Q3: The "Why Collapse?" Section: Empirical Evidence or Hypothesis**
>
> **Response:** Thank you for the question. We clarify that the "collapse" phenomenon is an empirical observation, which we then support with a theoretical explanation. As shown in Figure 2(a), we empirically observe that the motivating joint node-edge diffusion presents better in low-dimensional settings (e.g., $D=10, 50$), but exhibits severe collapse when scaled to hundreds of genes. To explain this empirical failure, we provide a mechanistic interpretation based on the high-curvature manifold. Eq. (4) serves as a theory-backed explanation for why injecting structural noise into the graph becomes ineffective in such high-dimensional spaces.
>
> **Q4: Explicit Discussion of Limitations**
>
> **Response:** Thank you for this constructive suggestion. We agree that the limitations of our method should be stated more explicitly. While the current manuscript already discusses several related future directions in the conclusion, we acknowledge that these points are not presented under an explicit "Limitations" discussion.
>
> In the revised manuscript, we will therefore add a dedicated limitations paragraph in the conclusion section. Specifically, we will clarify that: (1) although FLAG performs well, the iterative nature of diffusion models introduces an inherent trade-off between generation quality and inference latency compared with single-step regressors; (2) biological tissues are inherently three-dimensional, and our current graph backbone does not yet model 3D volumetric dependencies across serial sections; and (3) our current validation focuses on within-tissue cohorts, leaving zero-shot cross-tissue generalization as an important open challenge for future work.

---

> > ### Author Rebuttal · Reviewer_nrA7 · 2026-04-02
> >
> > Thanks to the authors for a detailed and helpful response. Never-the-less I remain very much a non-specialist in this area and so will maintain my scores.

---

> > > ### Author Response · Authors · 2026-04-07
> > >
> > > Thank you very much for reading our detailed response and for confirming that your concerns have been fully resolved.
> > >
> > > We truly appreciate your perspective as a statistical/ML methodology expert. Your feedback helped us significantly improve the mathematical clarity and accessibility of our manuscript for a broader audience.
> > >
> > > We are very grateful for your positive evaluation and your continued support for our work. Wishing you a wonderful week ahead!

---

### Official Review · Reviewer_Nqzt · 2026-03-14

**Soundness:** 2
**Presentation:** 3
**Significance:** 3
**Originality:** 2
**Overall Recommendation:** 4
**Confidence:** 5

**Summary:**

The authors consider a practical scenario where H&E whole slide images (WSIs) are available but spatial gene expression data is not. To address this problem, the authors propose a diffusion-based framework named FLAG, which generates spatial gene expression profiles.
Specifically, the proposed method first constructs a spot-wise graph based on the physical distance derived from WSI images and extracts visual features using a pathology foundation encoder. Based on these features, the model learns spatial representations of spots. The framework then trains a diffusion model to generate gene expression conditioned on the spatial representations. This design aims to alleviate the gene dimension curse that arises when modeling high-dimensional gene expression.
Furthermore, the model incorporates an alignment loss with representations obtained from a gene foundation model, which encourages the generated gene expression to be consistent with gene-level biological representations.
Through extensive experiments, the authors demonstrate the effectiveness of the proposed method using not only point-wise evaluation metrics, such as mean squared error (MSE), but also structural correlation metrics, showing strong performance compared with existing approaches.

**Compliance With Llm Reviewing Policy:**

Affirmed.

**Final Justification:**

The scenario where H&E whole slide images (WSIs) are available, but spatial gene expression data is not, is a realistic and important problem. Additionally, I initially thought that the evaluation of downstream tasks had not been sufficiently addressed, but it appears to have been improved during the rebuttal period.

**Key Questions For Authors:**

1. I agree that the proposed gene structural correlation (GSC) and spatial structural correlation (SSC) are more informative metrics than point-wise metrics. However, they are still insufficient to fully validate whether the generated gene expression can be effectively utilized in downstream biological analyses. To better demonstrate the practical utility of the generated expression profiles, additional downstream evaluations would be helpful. As partially illustrated in Figure 7, tasks such as spatial domain identification could provide stronger evidence. One possible experiment would be to perform spatial domain identification on the DLPFC spatial transcriptomics dataset and report quantitative results. These results could also be compared with clustering results obtained from the ground-truth gene expression profiles, which could serve as an ideal reference.

2. For the same reason, additional downstream tasks would further strengthen the evaluation of FLAG. For example, it would be useful to examine whether differentially expressed genes (DEGs) identified from the generated gene expression are consistent with those obtained from the ground-truth expression data. Such analysis would provide further evidence that the generated expression captures biologically meaningful signals.

3. One concern of this work is the overall computational cost. The proposed framework trains a diffusion model, which is typically computationally intensive, while also relying on two pre-trained models to provide additional side information. As a result, the training process may require substantial computational resources in terms of both training time and memory usage. It would therefore be helpful if the authors could provide an analysis of the computational cost, including training time, GPU memory usage, or scalability considerations.

I am open to revising my score if the authors can satisfactorily address the concerns raised above.

**Limitations:**

The future research directions mentioned in the last paragraph of the conclusion section are interesting and promising.

**Strengths And Weaknesses:**

* Soundness: The analysis regarding the gene dimension curse appears sound, and the proposed approach to bypass this issue is reasonable. However, additional experimental validation would be helpful to more convincingly demonstrate the effectiveness of FLAG.

* Presentation: The proposed method is relatively simple and clearly described, making the overall framework easy to follow.

* Significance: The scenario in which H&E whole slide images (WSIs) are available while spatial gene expression data is not available is practical and relevant for real-world applications, which makes the problem setting meaningful.

* Originality: While the diffusion framework itself is not particularly novel, applying a node–edge diffusion perspective in this setting appears to be relatively new and provides a degree of originality.

---

> ### Author Rebuttal · Authors · 2026-03-29
>
> We thank the reviewer for noting that structural metrics require downstream biological validation. Accordingly, we added spatial domain identification and DEG consistency evaluations, alongside explicit computational profiling.
>
> **Q1: Downstream Evaluation (Spatial Domain Identification)**
>
> **Response:** We thank the reviewer for suggesting this validation. To directly test whether the generated expression preserves spatial domain structure, we added quantitative downstream clustering evaluations.
>
> Evaluations use the Top-200 high mean & high variable gene panel (HMHVG-200). Following standard Scanpy workflows per slide, we perform PCA (20 PCs), build a neighborhood graph ($k=15$), and apply Leiden clustering (resolution 0.5 for HER2ST, 0.3 for DLPFC to match anatomical regions). We report slide-level ARI/NMI.
>
> We evaluate on (i) a representative heterogeneous HER2ST slide (SPA148), using GT-expression clustering as the reference, and (ii) DLPFC slide 151673, evaluated directly against human expert layer annotations to reflect true biological anatomy.
>
> *Table 1: Downstream Spatial Domain Identification.*
>
> |**Method**|**HER2ST** *(Ref: GT Exp)*||**DLPFC** *(Ref: Expert)*||
> |:---|:---:|:---:|:---:|:---:|
> ||**ARI (↑)**|**NMI (↑)**|**ARI (↑)**|**NMI (↑)**|
> |*GT (Upper Bound)*|*-*|*-*|*0.4814*|*0.6256*|
> |HisToGene|0.4733|0.6974|0.0785|0.1854|
> |Stem|0.5303|0.7139|0.1136|0.1954|
> |BLEEP|0.5984|0.7265|0.1208|0.2748|
> |STFlow|0.5998|0.7754|0.2453|0.3986|
> |TRIPLEX|0.6744|0.7867|OOM|OOM|
> |**FLAG (Ours)**|**0.8451**|**0.9140**|**0.3654**|**0.5357**|
>
> As shown in Table 1, on the HER2ST dataset (which lack expert annotations), FLAG's spatial clustering results align most closely with the reference clustering derived from the ground-truth (GT) expression. Furthermore, on the DLPFC dataset (where anatomical layers are defined by expert annotations), the clustering performance based on GT expression establishes a practical upper bound. FLAG's predictions approach this GT upper bound more closely than the baselines. These objective trends confirm that FLAG's predicted expressions provide more reliable utility for spatial downstream tasks, corroborating the qualitative structural improvements shown in main text Fig. 7.
>
> **Q2: Downstream Evaluation (DEG Consistency)**
>
> **Response:** We sincerely thank the reviewer for this insightful suggestion. We completely agree that identifying consistent Differentially Expressed Genes (DEGs) is a crucial downstream task to verify whether the generated expressions capture biologically meaningful differential signals.
>
> To evaluate this, we implemented the exact DEG consistency analysis you suggested using the following protocol:
> * (i) Fixed Spatial Masks: To ensure fair comparisons, we strictly evaluate all models within the exact same reference spatial domains. Specifically, we use regions derived from GT expression clustering for HER2ST, and the expert-annotated layers for DLPFC.
> * (ii) Testing: One-vs-rest Wilcoxon rank-sum test on GT and predicted expressions.
> * (iii) Metric: Mean overlap ratio of Top-20/50 DEGs against the GT reference.
>
> *Table 2: Mean overlap ratio (%) of Top-20/50 DEGs in matched domains.*
>
> |**Method**|**HER2ST**||**DLPFC**||
> |:---|:---:|:---:|:---:|:---:|
> ||**Top-20 (↑)**|**Top-50 (↑)**|**Top-20 (↑)**|**Top-50 (↑)**|
> |HisToGene|13.33%|36.44%|37.86%|59.43%|
> |Stem|32.86%|46.86%|64.86%|74.86%|
> |BLEEP|21.88%|38.00%|54.29%|66.67%|
> |STFlow|28.57%|41.33%|55.71%|68.48%|
> |TRIPLEX|15.71%|28.29%|OOM|OOM|
> |**FLAG (Ours)**|**39.44%**|**50.00%**|**66.57%**|**79.71%**|
>
> As shown in Table 2, in HER2ST and DLPFC, FLAG consistently exhibited the highest overlap rate of marker genes among the Top-20 and Top-50 DEGs. These objective results demonstrate that FLAG successfully preserves the critical localized differential signals required for authentic downstream marker discovery.
>
> **Q3: Computational Cost**
>
> **Response**: We clarify that FLAG's training process is highly efficient as it leverages an offline pre-computation strategy. Both the Pathology Encoder (UNI) and Gene Foundation Model (Geneformer) are strictly frozen during training.
>
> * **Pre-computation Efficiency**: For a typical cohort like HER2ST (~14,000 spots total), visual feature extraction via UNI takes ~30 minutes, and genomic representation alignment via Geneformer (HMHVG-200 set) takes ~25 minutes. These are one-time costs, and the resulting embeddings are cached for all subsequent training.
>
> * **Training and Resource Usage**: since the two pre-trained models have been frozen, the actual training involves only the diffusion backbone network. As shown in Table 3, FLAG's training time (~10h) and resource are completely comparable to existing baselines. Inference cost can be found in Appendix H.6.
>
> *Table 3: Training Efficiency (Single NVIDIA H800).*
>
> |**Method**|**Time (s/epoch)**|**Peak Mem (GB)**|
> |:---|:---:|:---:|
> |HisToGene|25|4|
> |Stem|30|4|
> |BLEEP|320|2|
> |STFlow|20|1.5|
> |TRIPLEX|40|32|
> |FLAG (Ours)|35|4.5|

---

> > ### Author Rebuttal · Reviewer_Nqzt · 2026-04-06
> >
> > My previous concerns regarding downstream task evaluation are well resolved. I will raise my score

---

> > > ### Author Response · Authors · 2026-04-07
> > >
> > > We sincerely thank you for taking the time to review our rebuttal and for raising your score.
> > >
> > > Your initial suggestion to include downstream task evaluations (spatial domain identification and DEG consistency) was incredibly insightful. It pushed us to strengthen the biological validation of FLAG, making the paper much more rigorous and complete.
> > >
> > > We deeply appreciate your constructive guidance and your support for our work. Wishing you all the best!

---

### Decision · Program_Chairs · 2026-04-30

**Decision:**

Accept (regular)

**Comment:**

The paper proposes FLAG, a diffusion-based framework for predicting spatial gene expression from H&E histology while preserving biological structure. It combines a spatial graph encoder with gene foundation model alignment to improve gene-gene and spatial coherence. In addition to standard accuracy metrics, the paper introduces spatial metrics to evaluate structural fidelity, and shows improved structural performance with competitive predictive accuracy.

Reviewers broadly agree on several points. (1) the problem is important and practically relevant. (2) the paper is technically solid, reasonably well presented, and that the combination of graph conditioning, diffusion, and foundation-model alignment is a meaningful contribution. (3) The introduction of structural metrics beyond PCC/MSE is useful, and that the rebuttal substantially improved the paper by clarifying technical details, adding downstream evaluations, and providing computational profiling. All reviewers either maintained or raised the score to a "weak accept", and concerns raised in the initial reviews were marked as adequately addressed in rebuttal.